



Global Observations and Modeling of Atmosphere-Surface Exchange of Elemental
Mercury – A Critical Review
W. Zhu[1,2], C.-J. Lin[1,3,*], X. Wang[1], J. Sommar[1], X. W. Fu[1], X. Feng[1,*]
[1] State Key Laboratory of Environmental Geochemistry, Institute of Geochemistry, Chinese Academy of
Sciences, Guiyang 550002, China
[2] Department of Chemistry, Umeå University, SE-901 87 Umeå, Sweden
[3] Center for Advances in Water and Air Quality, Lamar University, Beaumont, Texas 77710, United States
*Correspondence to:* C.-J. Lin (jerry.lin@lamar.edu) and X. Feng (fengxinbin@vip.skleg.cn)
C.-J. Lin, phone: +1 409 880 8761; fax: +1 409 880 8121; e-mail: jerry.lin@lamar.edu
X. Feng, phone: +86 851 85895728, fax: +86 851 85895095, e-mail: fengxinbin@vip.skleg.cn
*Manuscript for the Special issue of "Data collection, analysis and application of speciated atmospheric mercury"*
*in Atmospheric Chemistry and Physics (2015)*


**Abstract**
Reliable quantification of air-surfaces flux of elemental Hg vapor ($Hg^0$) is crucial for understanding
mercury (Hg) global biogeochemical cycles. There have been extensive measurements and modeling efforts
devoting to estimating the exchange fluxes between the atmosphere and various surfaces (e.g., soil, canopies,
water, snow, etc.) in past three decades. However, large uncertainty remains due to the complexity of $Hg^0$
bi-directional exchange, limitations of flux quantification techniques and challenges in model parameterization.
In this study, we provide a comprehensive review on the state of science in the atmosphere-surface exchange of
$Hg^0$. Specifically, the advancement of flux quantification techniques, mechanisms in driving the air-surfaces Hg
exchange, and modeling efforts are presented. Due to the semi-volatile nature of $Hg^0$ and redox transformation of
Hg in environmental media, Hg deposition and evasion are influenced by multiple environmental variables
including seasonality, vegetative coverage and its life cycle, temperature, light, moisture, atmospheric
turbulence, presence of reactants (e.g., $O_3$, radicals, etc.) that drives the physicochemical process of Hg in the
media where $Hg^0$ exchange occurs. However, effects of these processes on flux have not been fundamentally and
quantitatively determined, which limits the accuracy of flux modeling.
In this study, we compile an up-to-date global observational flux database and discuss the implication of
flux data on global Hg budget. Mean $Hg^0$ flux obtained by micrometeorological measurement did not appear to
be significantly greater than the flux measured by dynamic flux chamber methods over unpolluted surfaces
($p$=0.16, one-tailed, Mann-Whitney $U$ test). The spatio-temporal coverage of existing $Hg^0$ flux measurements is
highly heterogeneous with large data gaps existing in multiple continents (Africa, South Asia, Middle East,
South America and Australia). The magnitude of evasion flux is strongly enhanced by human activities,
particularly at contaminated sites. $Hg^0$ flux observations in East Asia are comparatively larger in magnitude than
the rest of the world, suggesting substantial reemission of previously deposited mercury from anthropogenic
sources. $Hg^0$ exchange over pristine surfaces (e.g., background soil and water) and vegetation need better
constrains for global analysis of atmospheric Hg budget. The existing knowledge gap and the associated research
needs for future measurements and modeling efforts for the air-surface exchange of $Hg^0$ are discussed.





## 1. Introduction

Mercury (Hg) is a global pollutant of broad concerns due to its toxicity, bioaccumulation characteristics and

adverse health effects (Driscoll et al., 2013), especially in its methylated forms such as monomethyl-mercury

($CH_3Hg$) species and dimethyl-mercury (($CH_3$)$_2$Hg) (Clarkson and Magos, 2006). Fish consumption has been

identified as the primary pathway for human exposure to $CH_3Hg$ (Mergler et al., 2007; Mason et al., 2012), while

the exposure through rice cultivated in areas with Hg pollution (e.g. mining and smelting areas) also poses a risk

(Feng et al., 2008a; Zhang et al., 2010). To protect human health and the environment from the adverse effects of

mercury, a global treaty "Minamata Convention for Mercury" that regulates Hg emission reduction from

anthropogenic sources has been signed by 128 countries since October 2013 (UNEP Minamata Convention,

2014). Emission of Hg into the atmosphere occurs from both natural processes and human activities. The release

of Hg from natural surfaces has been estimated to account for two thirds of global emissions (Fig. 1). However,

this estimate is subject to large uncertainty because of the challenges in quantifying the flux and in understanding

the mechanisms involved in the exchange process of elemental mercury vapor ($Hg^0$) (Selin, 2009; Zhang et al.,

2009; Gustin, 2011; Zhang et al., 2012a).

Hg emitted from the anthropogenic sources include all atmospheric species: gaseous elemental Hg ($Hg^0$,

GEM), gaseous oxidized Hg (GOM), and particulate bound Hg (PBM) (Pacyna et al., 2006; AMAP/UNEP,

2013), while evasion derived from the Earth's surfaces is dominated by GEM (Gustin, 2011). Owing to the high

deposition velocity of GOM and PBM (~1-2 orders higher than GEM) (Zhang et al., 2009), GOM and PBM are

readily deposited locally and regionally while GEM is subject to long-range transport (e.g., hemisphere scale)

and can deposit remotely from the emission sources (Lindberg et al., 2007; Gustin and Jaffe, 2010). Atmospheric

Hg continuously goes through the deposition and re-emission cycle while undergoing physical and chemical

transformations (Lin and Pehkonen, 1999).

Extensive efforts have been devoted to understanding the spatial and temporal pattern of $Hg^0$ exchange flux.

Geogenically Hg-enriched surfaces and anthropogenically polluted sites are strong Hg emission sources

(Kocman et al., 2013). Emissions from natural sources and from previously deposited $Hg^0$ on substrate surfaces

are not analytically distinguishable using current measurement techniques (cf. Section 2). Direct measurement of

$Hg^0$ flux from the background surfaces is difficult due to small vertical $Hg^0$ concentration gradient (therefore low





flux) (Zhu et al., 2015a). Since the first application of a stainless steel dynamic flux chamber for $Hg^0$ flux
measurement over background lakes and soils in 1980s (Schroeder et al., 1989; Xiao et al., 1991), significant
advancement in the experimental approaches (e.g., dynamic flux chamber, micrometeorological methods,
$Hg^0/^{222}Rn$ flux ratio, enriched isotope tracer methods, open-path laser optical spectroscopic method, and $Hg^0/CO$
ratio) have been made (Sommar et al., 2013a). However, a standard protocol for $Hg^0$ flux quantification does not
exist (Gustin, 2011; Zhu et al., 2015b), which complicates the comparison and interpretation of flux data
reported in the literature (cf. Section 4).
In this study, we present a comprehensive review on the global observation of $Hg^0$ flux in the peer-reviewed
literatures, and provide a state-of-the-science assessment on the air-surface exchange of $Hg^0$. Specifically, the
advancement of flux quantification techniques, physicochemical factors driving the exchange process, existing
field data of $Hg^0$ flux, and modeling efforts for scaling up the measured flux for global assessment are
synthesized. Furthermore, the spatial and temporal characteristics of $Hg^0$ flux, as well as the underlying
influencing factors are investigated. Key knowledge gaps, future directions for field measurements, and
development of new-generation air-surface exchange model for $Hg^0$ flux are discussed.

## 85  2. Advances in $Hg^0$ flux quantification methods

The theory and application of $Hg^0$ flux measurement techniques have been documented extensively (Zhang
et al., 2009; Gustin, 2011; Sommar et al., 2013a). Here we focus on the developments, advantages and
disadvantages, and comparability and uncertainties of different flux quantification techniques. DFCs,
micrometeorological techniques (MM), and bulk methods (e.g., $Hg^0/^{222}Rn$ flux ratio, enriched isotope tracers)
are the mostly widely applied approaches for surface-atmosphere $Hg^0$ flux quantification (Schroeder et al., 1989;
Xiao et al., 1991; Kim and Lindberg, 1995; Kim et al., 1995; Cobos et al., 2002; Amyot et al., 2004; Olofsson et
al., 2005; Obrist et al., 2006; Bash and Miller, 2008; Lin et al., 2012; Slemr et al., 2013; Zhu et al., 2013c), of
which DFCs and MM techniques accounted for >95% of all observations documented to date (cf. Section 4).
Open-path laser optical spectroscopic (LIDAR) method and $Hg^0/CO$ ratio were applied to estimate Hg emission
from area/regional sources (e.g. LIDAR: mining areas, industrial plants, geothermal sites; $Hg^0/CO$ ratio:
continental level atmospheric Hg transport) (Aldén et al., 1982; Edner et al., 1991; Sjöholm et al., 2004; Jaffe et



al., 2005; Fu et al., 2015a). There has not been a standardized protocol for any of the techniques (e.g.,
instrumentation set-up, operation parameters) (Gustin, 2011; Zhu et al., 2015b). Recent collocated
measurements and uncertainties analysis emphasized the importance of method standardization and processing
of field data acquired by the measurement systems (Fritsche et al., 2008b; Converse et al., 2010; Zhu et al.,
2015a). Application of appropriate flux measurement technique depends on the scalar detection accuracy, sensor
response frequency and level of automation (Sutton et al., 2007). The traditional standard procedure of sampling
ambient air $Hg^0$ is by enhancement collection onto traps containing gold (Fitzgerald and Gill, 1979; Slemr et al.,
1979). A wide-spread continuous $Hg^0$ monitor is the automated dual channel, single amalgamation, cold vapor
atomic fluorescence analyzer (Model 2537, Tekran Instruments Corp.), which relying on this principle. The
certified detection limit is < 0.1 ng m$^{-3}$. However, the pre-concentration procedure takes ≥2.5 min and therefore
real-time, high-frequency data acquisition is not possible (Gustin, 2011; Fu et al., 2012b; Gustin et al., 2013;
Gustin et al., 2015). Monitoring ambient air $Hg^0$ with a higher frequency (≤ 1Hz) can be achieved by using
Lumex RA-915+ Zeeman atomic absorption spectrometry (AAS) analyzer operating without trap
pre-concentration. However, the instrument has a detection limit ~1 ng m$^{-3}$ and therefore is preferred for
industrial level studies but applicable under ambient $Hg^0$ concentration (Holland K., 2005). More recently, high
frequency (25 Hz) cavity ring-down spectroscopy (CRDS) sensor has been deployed for $Hg^0$ concentration
measurement, but it has a higher detection limit (> 0.35 ng m$^{-3}$) and suffers from sensor's baseline drifting and
interferences with $O_3$ (Fain et al., 2010; Pierce et al., 2013). Another laser technique, laser-induced fluorescence
sensor, has been designed and successfully applied for up to one day continuous measurement with improved
detection limit (~15 pg m$^{-3}$) (Bauer et al., 2002; Bauer et al., 2014). However, both methods have not been yet
proved to apply for long-term field measurement. The coupling with a commercial instrument (e.g. Tekran®
2537) renders continuous and unattended flux measurements by DFC or MM techniques to be accomplished and
are most widely deployed over various surfaces (cf. Section 4). However, this implementation is associated with
a significant cost, for which the expense of the $Hg^0$ analyzer is normally exciding that of the essential flux
system.

**2.1 Dynamic flux chambers**



The DFC method (footprint generally <0.1 m$^2$) is a frequently used Hg$^0$ flux measurement technique over
soils, water surfaces, and low-stand grass due to its relatively low cost, portability, and versatility (Sommar et al.,
2013a). DFCs operating under steady-state (Xiao et al., 1991; Carpi and Lindberg, 1998) and non-steady-state
conditions (Rinklebe et al., 2009) are used in Hg research with the former configuration by far the most common.
Dynamic flux bag (DFB) has been applied for flux measurement over tall grass and tree branches (Zhang et al.,
2005; Graydon et al., 2006; Poissant et al., 2008). Laboratory mesocosms probing whole ecosystem Hg
exchange have also been attempted; a 180×10$^3$ L chamber (7.3 × 5.5 × 4.5 m$^3$) was deployed for quantifying
soil-plant-atmosphere flux (Gustin et al., 2004; Obrist et al., 2005; Stamenkovic and Gustin, 2007; Stamenkovic
et al., 2008). Construction materials such as fluorinated ethylene propylene (FEP) films and quartz have been
recommended for DFCs due to its high actinic light transmittance and low blank (Kim and Lindberg, 1995; Carpi
et al., 2007; Lin et al., 2012). DFCs volumes and flushing flow rates reported ranged from 1 to 32 L and 1.5 to 20
L min$^{-1}$, resulting a turnover time (TOT) ranging from 0.1 to 14 min (Eckley et al., 2010; Zhu et al., 2011). Using
DFCs, Hg$^0$ flux is calculated as:
$$F = \frac{Q(C_{out} - C_{in})}{A} \qquad\qquad (1)$$
where $F$ is Hg$^0$ flux (ng m$^{-2}$ h$^{-1}$), $Q$ is DFC internal flushing flow rate (m$^{-3}$ h$^{-1}$), $A$ is DFC footprint, $C_{out}$ and
$C_{in}$ are the Hg$^0$ concentrations at the DFC outlet and inlet, respectively. Eq. (1) relies on mass balance
calculation of two $C_{out}$ and two $C_{in}$ measurements alternately and assumes that the surface shear velocity over
the DFC footprint is uniform and therefore results in a constant flux spatially over the wetted surface. Distinct
Hg$^0$ fluxes have been observed using DFCs of different designed shapes under similar environmental conditions
(Eckley et al., 2010). Lin et al. (2012) investigated the internal flow field and Hg$^0$ concentration distribution in
two commonly designed DFC (i.e. rectangular and dome-shaped chambers) showed that the airstream inside the
DFCs is not uniform and the surface shear flow is divergent over the footprint, resulting in a non-uniform Hg$^0$
concentration gradient over the substrate surface. Eckley et al. (2010) systematically investigated effects of
fabrication material, footprint, chamber dimensions including port positions and flushing flow rates on the
measured Hg$^0$ flux by DFCs. Consistent with previous studies, flushing flow rate is among the most influential





factor that if varied may induce up to one order of magnitude differences in the observed fluxes (Gustin et al.,
1999; Wallschläger et al., 1999; Gillis and Miller, 2000a; Lindberg et al., 2002c; Zhang et al., 2002).
Computational fluid dynamic modeling of DFC mass transfer indicated that smaller diffusion resistance at
higher flushing flow rate yielded higher measured flux. However, due to the non-uniform internal $Hg^0$
concentration gradient, measured $Hg^0$ flux from substrates may change unpredictably when flushing flow rate
varies (Eckley et al., 2010; Lin et al., 2012), which should be taken into consideration when flux obtained by
DFCs of different designs and flushing flow rates cannot be directly compared. Another limitation of DFCs is the
isolation of chamber internal surfaces from ambient condition. This excludes the effect of atmospheric
turbulence and therefore may cause a large uncertainty when using DFC data as in-put for scale-up estimation.
Lin et al. (2012) proposed an aerodynamic designed chamber (NDFC) which enables producing a uniform
surface friction velocity to link with ambient shear condition to rescale to the ambient flux using Eq. (2), which
allows to utilize ambient surface shear condition rather than artificial steady flushing flow rate to calculate flux:
$$F = \frac{Q(C_{out} - C_{in})}{A} \cdot \frac{k_{mass(a)}}{k_{mass(DFC)}} \qquad (2)$$

where $k_{mass(a)}$ is the overall mass transfer coefficient under ambient condition, and $k_{mass(DFC)}$ is overall mass
transfer coefficient in the DFC measurement area.

In addition to the uncertainties caused by varying flushing flow rates, altered short and long wave radiation

balance within DFCs resulting in a modified micro-environment were found to bias the observed flux (Zhu et al.,
2015a). DFC flux is measured through intermittent sampling of ambient and chamber air for $Hg^0$ analysis using
a single detector (Lindberg et al., 2002c), which assumed that ambient $Hg^0$ variability was negligible during air
sampling. At locations where significant variation in $Hg^0$ concentration exist (e.g., sites with anthropogenic
emission sources), Eckley et al. (2011a) proposed a data assimilation protocol: $|\Delta C_{oi}| > |\Delta C_{ii}|$ should be valid
for each calculated flux, otherwise the flux should be rejected ($\Delta C_{oi}$ is the difference between $C_{out}$ and the
average of two $C_{in}$ which before and after taking $C_{out}$, while $\Delta C_{ii}$ is the difference between above two $C_{in}$).
The concern about influencing plant physiology restricts the deployment of small DFCs to short term field





measurements over the same vegetated plot. Given the small footprint and that $Hg^0$ fluxes over terrestrial
surfaces are profoundly variable in space and time, replication DFC measurements are thus preferred but often
not carried out.

## 177   2.2 Micrometeorological methods

MM methods differ in measurement principles and spatial scale of flux footprint compared to DFCs and
have the capability of measuring ecosystem-scale (typically hectare scale) flux under undisturbed conditions and
represent a preferred flux quantification techniques over vegetated landscapes. MM techniques for background
$Hg^0$ flux measurements currently comprises of relaxed eddy accumulation method (REA), aerodynamic gradient
method (AGM), and modified Bowen-ratio method (MBR). The preferred MM technique, eddy covariance (EC),
a direct flux measurement method without any applications of empirical constants, requires a fast response (~10
Hz) gas analyzer, and has not been realized for regular $Hg^0$ flux measurements (Aubinet et al., 2012). Recently,
Pierce et al. (2015) reported the first field trial of CRDS-EC flux measurement over Hg enriched soils with a
minimum flux detection limit of 32 ng m$^{-2}$ h$^{-1}$, insufficient for $Hg^0$ flux measurement at most, if not all,
background sites. Sommar et al. (2013a) and Zhu et al. (2015b) detailed the theory, computation, and existing
MM approaches for measuring $Hg^0$ fluxes. Gradient methods rely on quantifying the vertical concentration
gradient (two or more heights sampling), turbulent parameters (AGM) or scalar concentration gradient (MBR),
and scalar EC-fluxes. A major advantage of REA method is that REA up- and down-draft sampling conducted at
one height, which overcomes the uncertainties associated with: (1) footprint differences due to two heights
sampling in gradient methods, and (2) possible oxidation/reduction introduced forming or loss of $Hg^0$ between
the two heights. On the other hand, the analytical requirement for REA is more stringent than for the gradient
methods, especially under windy conditions, increasing the demand on the precision of the sampling and
chemical analysis (Zhu et al., 2015a).
REA method has been deployed for flux measurement for agricultural lands, forest canopies, wetlands, and
urban settings (Cobos et al., 2002; Olofsson et al., 2005; Bash and Miller, 2009; Osterwalder et al., 2015;
Sommar et al., 2013b). AGM method has been used over grasslands, agricultural lands, saltmarsh, landfills, and
snow (Lee et al., 2000; Kim et al., 2001; Kim et al., 2003; Cobbett et al., 2007; Cobbett and Van Heyst, 2007;



Fritsche et al., 2008c; Fritsche et al., 2008b; Baya and Van Heyst, 2010). MBR method has been set up in
grasslands, forest floor, agricultural lands, lakes, wetlands, and snow (Lindberg et al., 1992; Lindberg et al.,
1995b; Lindberg et al., 1998; Lindberg and Meyers, 2001; Lindberg et al., 2002b; Brooks et al., 2006; Fritsche et
al., 2008b; Converse et al., 2010). The theoretical and application requirement of micrometeorology is less
restricted for large areas of uniform vegetation (or soil) in flat landscapes, where an atmospheric surface layer
develops and the horizontal flux variability is low in the absence of pollution plumes and the flux above the
surface remains constant with height (Wesely and Hicks, 2000). Under these turbulent exchange conditions, the
flux acquired at the measurement height resembles the actual flux at the surfaces under measurement. There are
several potential causes that can invalidate the above assumptions. For instance, the advection of $Hg^0$ from the
nearby sources to the measurement site may occur. It is known that local point sources of $Hg^0$ can affect MM
measurements downwind (Bash and Miller, 2007). Loubet et al. (2009) estimated such advection errors in $NH_3$
gradient flux to result in 2.1% to 52% of vertical flux at a monitoring site at 810m downwind of $NH_3$ source (a
farm building) implying a significant error contribution from advection. Large variation of $Hg^0$ fluxes measured
by MBR methods were also reported at nighttime as a result of advection in Nevada STORM project, however,
the error have not been quantified (Gustin et al., 1999). For $Hg^0$ flux over forest canopies, the influence of
within-canopy source and sink terms on net ecosystem flux has not been evaluated. A multiple heights
gradients/REA measurements is needed to resolve the true flux. Since there is not a reliable sensitive $Hg^0$ sensor
at high measurement frequency, an empirical multiplication factor or proxy scalar is required for computing all
MM-$Hg^0$ flux (e.g., relaxation coefficient $\beta$ derived from a selected proxy scalar for REA, eddy diffusivity
$K_H$ derived from sensible heat for AGM, and proxy scalar such as sensible heat, $CO_2$, and $H_2O$ flux for MBR)
(Lindberg et al., 1995a; Edwards et al., 2005; Baya and Van Heyst, 2010; Zhu et al., 2015b). These empirical
factors may introduce uncertainties when the proxy scalar value is small, which frequently occur during dawn
and dusk and under the condition of low atmospheric turbulence. Proxy scalar inferred relaxation coefficient
( $\beta_{CO_2}$, $\beta_{T_s}$, $\beta_{H_2O}$ ) is typically not significantly different (~0.56) during a campaign above wheat agricultural
land, while all $\beta$ values were highly variable when the corresponding scalar flux was close to zero (Gronholm
et al., 2008; Sommar et al., 2013b). Converse et al. (2010) reported $Hg^0$ flux over a wetland meadow using



collocated AGM and MBR methods for four campaigns during an entire year. They found comparable fluxes in
summer, while source/sink characteristics reversed between the two methods in fall and winter. Zhu et al. (2015b)
found that AGM and MBR observed similar $Hg^0$ fluxes when absolute sensible heat flux was >20 W $m^{-2}$; and the
agreement is not satisfactory when the absolute sensible heat flux was <20 W $m^{-2}$. Rejecting flux data collected
under low turbulence conditions can bias the integrated flux over time (Mauder and Foken, 2004); and adequate
data rejection and correction approaches need to be developed (Aubinet et al., 2012).

**2.3 Comparability of flux measured by micrometeorological and chamber methods**
Limited efforts have been devoted to understand the flux disparity caused by different flux measurement
techniques. The Nevada STORMS project was the first attempt using eleven collocated measurements (7 DFC
methods and 4 gradient-based MM methods) to simultaneously quantify $Hg^0$ flux from Hg-enriched bare soils in
September 1997 (Gustin et al., 1999; Lindberg et al., 1999; Poissant et al., 1999; Wallschläger et al., 1999). In the
campaign, the mean fluxes obtained using MM methods were three times greater than those obtained by DFCs
(Fig. 2a). One possible reason for the low observed flux by DFC was the small flushing flow rates
(corresponding TOT: 1.1-24 min) that were not sufficient to eliminate the accumulated $Hg^0$ in the DFC and
subsequently suppressed $Hg^0$ evasion. Later, Gustin and coworkers extended the study at the same site using a
1-L polycarbonate DFC (TOT: 0.2 min) (Engle et al., 2001) and a MBR method (Gustin et al., 1999) in October
1998 (Gustin, 2011). Although MBR show substantial flux variability, DFC and MM fluxes were not
significantly different ($p$>0.05) for dry and wet diel flux cycles (Fig. 2b). Two challenges in comparing MM and
DFCs fluxes in these studies were the site heterogeneity (1.2-14.6 µg Hg $g^{-1}$ in soil) and the footprint differences.
The footprint of MM methods was estimated to be 40-70 m upwind the sampling sites (50-200 $m^2$) while DFC
covered only 0.12-0.3 $m^2$ (Gustin et al., 1999). Recently, an integrated field $Hg^0$ flux methods intercomparison
project measured $Hg^0$ flux from a background homogenized agricultural field (~45 ng Hg $g^{-1}$) using REA, AGM,
MBR, a polycarbonate NDFC (TOT: 0.47 min), and a traditional quartz DFC (TDFC, TOT: 0.32 min) (Fu et al.,
2008a; Zhu et al., 2015a, b). Overall, MM fluxes showed highly dynamic temporal variability while DFCs
followed a gradual diel cycle similar to those temperature and solar irradiance. REA observed a broader flux
distribution similar to $NH_3$ and $CH_4$ fluxes observed by MM techniques (Beverland et al., 1996; Moncrieff et al.,



1998; Nemitz et al., 2001). The median fluxes obtained by REA, AGM, and MBR were not significantly
different (Friedman two-way analysis, $\chi^2 = 1.29 < \chi^2_{p=0.05} = 5.99$). Over a three-week period, NDFC obtained a
comparable mean flux with AGM and MBR which are approximately three times of the TDFC flux, implying
that NDFC potentially reduced uncertainty using real atmospheric boundary shear condition to rescale (Lin et al.,
2012). However, the correlation between NDFC/TDFC and MBR flux are weak because of high variability of
MM flux (Fig. 2c). Pierce et al. (2015) observed comparable mean flux from simultaneous measurement using
CRDS-EC, MBR, and DFC (849 ng m$^{-2}$ h$^{-1}$, 1309 ng m$^{-2}$ h$^{-1}$, and 1105 ng m$^{-2}$ h$^{-1}$, respectively) over Hg-enriched
soils, similar flux patterns were recorded from CRDS-EC and MBR.

Fig. 3 showed the comparisons of Hg$^0$ fluxes measured by MM methods and DFCs from relative

homogeneous landscapes reported in the literature (cf. Section 4, substrate total Hg < 0.3 μg Hg g$^{-1}$). MM
methods yield a broader Hg$^0$ flux range compared to DFCs methods, consistent with the field campaigns using
collocated measurements (Zhu et al., 2015b). MM mean flux is higher than DFCs flux by a factor of two
approximately, which may be a result by the fact that a large fraction of DFC measurements utilized a relative
low TOT underestimating surface flux. However, Mann-Whitney $U$ test indicated that the differences between
the two methods are not significant ($p$=0.16, one-tailed). Probability of the two data sets showed positive
skewness (4.2 and 3.9 for MM and DFCs, respectively) and kurtosis (19.6 and 27.2) caused by those high flux
observations, likely resulting from asymmetrical data distribution as well as the differences in measurement site
and periods. The flux data of MM methods in Fig. 3 were mostly obtained from agricultural fields (33%) and
grasslands (36%) while the data of DFC methods were mainly from background sites (68%); and MM
measurement generally covered a longer period (weeks to year) compared DFC measurements lasted a much
shorter period (hours, days to a few weeks). Typically, significant Hg$^0$ evasion is observed during daytime, while
deposition, bi-directional exchange, or mild emission occurs at nighttime (cf. Section 4). Agnan et al. (2015)
summarized MM and DFC fluxes observed in laboratory and during field campaigns over terrestrial substrates,
and found that observed the median MM flux (-0.01 ng m$^{-2}$ h$^{-1}$, n=51) was statistically smaller than the median
DFC flux (0.5 ng m$^{-2}$ h$^{-1}$ and 1.75 ng m$^{-2}$ h$^{-1}$ for flushing flow rate ≤ 2 L min$^{-1}$ and > 2 L min$^{-1}$, $p < 0.05$). They
suggest elevated flushing flow rate generated partial vacuum inside DFC created artificial Hg$^0$ flux from soil
even at <2 L min$^{-1}$, although this is not supported by the large Hg$^0$ concentration gradient (inside and outside





DFC) formed at low flushing flow rate (Zhang et al., 2002; Eckley et al., 2010). An alternative explanation is that
MM measurements were predominantly deployed for background vegetated surfaces while DFC were mainly
applied for soil surfaces, the difference in the source/sink characteristics over vegetation and bare soils may
cause the difference in median fluxes.

**3. Factors influencing air-surface $Hg^0$ exchange**
**3.1 Air-soil Hg exchange**
Meteorological parameters (solar radiation, soil/air temperature, atmospheric turbulence), soil substrate
characteristics (e.g. Hg content, soil moisture, organic matters, porosity, and microbial activity), and ambient air
characteristics (e.g. $Hg^0$ and $O_3$ concentration) can influence the air-surface exchange of $Hg^0$. Changes of these
factors force two controlling processes: (1) formation of evaporable $Hg^0$, and (2) mass transfer of $Hg^0$. Solar
radiation has been found highly positively correlated with soil $Hg^0$ flux (Carpi and Lindberg, 1997; Boudala et
al., 2000; Zhang et al., 2001; Gustin et al., 2002; Poissant et al., 2004a; Bahlmann et al., 2006), which is
generally regarded as enhancing $Hg^{II}$ reduction and therefore facilitating $Hg^0$ evasion (Gustin et al., 2002).
Actinic light spectral analysis suggested UV-B can reduce $Hg^{II}$ to $Hg^0$ over soil, while UV-A and visible light
have a much lower enhancement (Moore and Carpi, 2005; Choi and Holsen, 2009b). Temperature is an
important factor that promotes $Hg^0$ evasion, typically described by Arrhenius equation (Carpi and Lindberg,
1997; Poissant and Casimir, 1998; Gustin et al., 2002). However, Arrhenius relationship cannot explain $Hg^0$ flux
spikes at sub-zero temperatures, implying other mechanisms such as the expansion and contraction of liquid
fraction in soil substrates occurred (Corbett-Hains et al., 2012). Atmospheric turbulence (i.e. wind, surface
friction velocity) is another factor in driving the $Hg^0$ release from soil (Lindberg et al., 1999; Wallschläger et al.,
1999). Increased turbulence enhances $Hg^0$ mass transfer and promotes $Hg^0$ desorption from soil (Gustin et al.,
1997; Lindberg et al., 2002c; Zhang et al., 2002; Eckley et al., 2010; Lin et al., 2012).
Soil types, soil moisture and Hg content in soil are also important factors influencing observed $Hg^0$ flux (Xu
et al., 1999; Kocman and Horvat, 2010; Lin et al., 2010a). Lindberg et al. (1999) observed that rainfall and
irrigation enhances soil $Hg^0$ emission by an order of magnitude. Subsequent studies supported that adding water
to dry soil promotes Hg reduction and that water molecular likely replaces soil $Hg^0$ binding sites and facilitates



Hg$^0$ emission. In saturated soil, Hg emission is suppressed because the soil pore space is filled with water, which
hampers Hg mass transfer (Gillis and Miller, 2000b; Gustin and Stamenkovic, 2005). Pannu et al. (2014)
investigated Hg$^0$ flux over boreal soil by manipulating soil moisture, maximum flux was observed at 60% soil
moisture (water filled pore space), whereas flux become inhibited at 80%. Repeated rewetting experiments
showed smaller increase in emission, implying "volatizable" Hg$^0$ needs to be resupplied by means of reduction
and dry deposition after a wetting event (Gustin and Stamenkovic, 2005; Song and Van Heyst, 2005; Eckley et al.,
2011b). Soil organic matter (SOM) have a strong affinity with Hg$^0$ and form stable complexes with Hg$^{II}$ (Grigal,
2003; Skyllberg et al., 2006), and therefore diminish soil Hg$^0$ efflux (Yang et al., 2007). Mauclair et al. (2008)
measured Hg$^0$ flux from sand (0.5 µg Hg g$^{-1}$) spiked with humic substances; and found that Hg$^0$ flux decreased
sharply by incremental addition of up to 0.1% of humic matter. Higher soil porosity has also been suggested to
facilitate Hg$^{II}$ reduction and Hg$^0$ transfer from soil (Fu et al., 2012a). Microbial induced reduction can enhance
Hg$^0$ evasion but to a less extent (Fritsche et al., 2008a; Choi and Holsen, 2009b). Higher flux has also been
observed by increasing soil pH value (Yang et al., 2007).

Elevated ambient Hg$^0$ concentration has been found to suppress Hg$^0$ flux by reducing Hg$^0$ concentration

gradient at the interfacial surfaces (Xin and Gustin, 2007). At locations where ambient Hg concentration is high
(e.g., mining sites, landfills), deposition is predominately observed despite of the influence of meteorological
factors (Bash and Miller, 2007; Wang et al., 2007b; Zhu et al., 2013c). Atmospheric O$_3$ was found to induce
not-yet-understood chemical processes that enhance Hg$^0$ emission from soil in the dark (Zhang et al., 2008).
Laboratory experiments showed that Hg$^0$ flux from soils with Hg$^{II}$ as the dominant species can be enhanced by
1.7 to 51 times in the presence of O$_3$ (50-70 ppb), and be decreased by >75% over Hg$^0$-amended soils (Engle et
al., 2005). Environmental factors interacts naturally (e.g., irradiation and temperature), which can impose
synergistic and antagonistic effects on forcing Hg$^0$ flux changes (Gustin and Stamenkovic, 2005). Fig. 4 shows
the individual effects and synergism between solar radiation, air temperature, and water content on Hg$^0$ flux from
a typical low organic content soil (~1.5 wt % ) (Lin et al., 2010a). All three individual factors enhance flux by
90%-140%, while two-factor synergetic effect accounts for 20%-30% enhancement.

**3.2 Air-vegetation Hg$^0$ exchange**





Vegetation alters air-ecosystem $Hg^0$ flux through (1) changing environmental variables at ground surfaces
(e.g., reducing solar radiation, temperature, and friction velocity) (Gustin et al., 2004), and (2) provide active
surface for Hg uptake. Carpi et al. (2014) reported forest floor soil fluxes of -0.73±1.84 and 0.33±0.09 ng $m^{-2}$ $h^{-1}$
from intact New England and Amazon forest floors, respectively. Substantial emission fluxes at 9.13±2.08,
21.2±0.35 ng $m^{-2}$ $h^{-1}$ were observed after deforestation suggested forest coverage effectively reduced ground
floor $Hg^0$ emission. More importantly, air-plant interaction increases the complexity of air-terrestrial Hg
exchanges; and the role of vegetation as a source or a sink of atmospheric Hg has been in debates in the literature.
Lindberg et al. (1998) observed a significant $Hg^0$ emission from forest canopies in Tennessee and Sweden
(10-300 and 1-4 ng $m^{-2}$ $h^{-1}$); estimated annual $Hg^0$ emission from global forest to be 800-2000 tons; and
emphasized the need for a re-assessment on this potentially important source. Based on the observed Hg
presence in xylem sap (Bishop et al., 1998), plant has been hypothesized as a conduit for releasing geospheric Hg
to the atmosphere (Leonard et al., 1998a, b). Subsequent models simply treated plant emission as a function of
evapotranspiration rate (Xu et al., 1999; Bash et al., 2004; Gbor et al., 2006; Shetty et al., 2008). However, recent
measurement suggested that air-surface exchange of $Hg^0$ is largely bidirectional between air and plant and that
growing plants act as a net sink (Ericksen et al., 2003; Stamenkovic et al., 2008; Hartman et al., 2009). Stable Hg
isotope tracer studies have shown that Hg in soils cannot be translocated from roots to leaf due to the transport
barrier at the root zone (Rutter et al., 2011b; Cui et al., 2014), suggesting that the source of Hg in leaf is of
atmospheric origin.
Hg concentration in foliage is generally influenced by the level of air $Hg^0$ present in air $Hg^0$ (Ericksen et al.,
2003; Frescholtz et al., 2003; Ericksen and Gustin, 2004; Millhollen et al., 2006a; Fay and Gustin, 2007a; Niu et
al., 2011). Climate factors (e.g., solar irradiation, temperature), biological factors (e.g., leaf age, plant species),
and ambient air components (e.g., $CO_2$) also significantly influence on foliar $Hg^0$ flux (Rea et al., 2002;
Millhollen et al., 2006a; Millhollen et al., 2006b; Fay and Gustin, 2007a; Bushey et al., 2008; Stamenkovic and
Gustin, 2009; Rutter et al., 2011a). For instance, higher Hg concentration found at the bottom aged leaf suggest
the influence of longer exposure time (Bushey et al., 2008) over an immediate source from soil (Frescholtz et al.,
2003). Stomatal and non-stomatal (e.g., cuticle) processes are both viable pathways for bidirectional Hg
exchange (Stamenkovic and Gustin, 2009). Stomatal process may play a predominant role as Hg accumulated on



cuticle surface was generally <10% of total Hg content in leaf (Rutter et al., 2011a; Laacouri et al., 2013). Solar
radiation, temperature, and $CO_2$ concentrations regulating plant stomatal activity may therefore affect Hg uptake
and gas exchange. For instance, high air-vegetation $Hg^0$ flux observed during daytime show deposition, opposite
to daytime evasion observed over other terrestrial surfaces (cf. Section 4.3) (Stamenkovic et al., 2008). In
addition, Hg in leaf has been shown to be assimilated into leaf biomass during the growing stage (Bash and
Miller, 2009), suggesting Hg uptake occurs with plant assimilation metabolism.

It has been proposed that Hg in leaf can be classified as two forms: (1) exchangeable Hg which can be

re-emitted back to the atmosphere, and (2) biological assimilated Hg retained in leaf (Rutter et al., 2011a).
However, whether or not $Hg^0$ can be oxidized after uptake into tissue, and the possibility of assimilated Hg being
reemitted from leaf (e.g., reduction of leaf retained $Hg^{II}$ or un-oxidized $Hg^0$ originally from ambient air) remain
unclear. Many studies observed a so-called "compensation point" denoting the interfacial concentration of Hg
that drive the concentration gradient for bi-directional air-vegetation exchange of $Hg^0$ (Hanson et al., 1995;
Poissant et al., 2008; Bash and Miller, 2009). However, the hypothesis of compensation point does not explain
the accumulation of Hg in vegetation pool. Recent Hg isotopic fractionation studies show promise for exploring
air-leaf Hg exchange mechanism. Demers et al. (2013) reported a kinetic mass dependent fractionation (MDF,
$\delta^{202}Hg$) of -2.89‰ during air-leaf Hg exchange from air to leaf. The result indicated that uptake of atmospheric
Hg by leaf occurs, and the deposited Hg is likely to be chemically bonded in leaf with sulfur and nitrogen
functional groups in enzymes within stomatal cavities (Rutter et al., 2011a), rather than with carboxylic ligands
on leaf surface. Another important finding is the negative mass independent fractionation (MIF, $\Delta^{199}Hg$) of Hg of
-0.19‰ to -0.29‰, correlated well with $Hg^{II}$ photochemical reduction by low molecular mass organic matter
with sulfur-containing ligands (Zheng and Hintelmann, 2010). This implies that the Hg reemission may result
from revolatilization of chemical bounded Hg in leaf. However, they did not rule out the potential influence of
PBM and GOM that deposit on the leaf, which may undergo partial uptake by plant with the remaining being
reemitted back to the atmosphere.

**3.3 Air-water Hg exchange**

Bulk method, DFC and MM methods have been utilized in air-water $Hg^0$ flux measurement. Bulk methods



is the most widely utilized approach for oceanic surface (>80% of the field data, Table 1). Sommar et al. (2013a)
summarized the methodologies of the bulk method, which generally controlled by both kinetic (overall mass
transfer coefficient, $k$) and thermodynamic (partial pressure related concentration gradients) forcing
(Wanninkhof, 1992; Wanninkhof et al., 2009; Kuss et al., 2009; Kuss, 2014):
$$F = k \times \left( DGM - GEM \middle/ H_T^{'} \right) = 0.31 \times U_{10}^2 \times \left( \frac{v}{600 \times D_{Hg^0}} \right)^{-0.5} \times \left( DGM - GEM \middle/ H_T^{'} \right) \qquad (3)$$

where DGM is dissolved gaseous Hg concentration in the surface water film, GEM is near surface gas $Hg^0$
concentration, $H_T^{'}$ is dimensions Henry's law constant, $U_{10}$ is wind speed at 10m, $v$ is the water kinematic
viscosity, and $D_{Hg^0}$ is $Hg^0$ diffusion coefficient in water. Fig. 5 shows air-surface exchange processes and
transformation of DGM in water phase. From a kinetic point of view, the overall mass transfer coefficient of $Hg^0$
is described by a molecular diffusivity in the water and gas film. Since the mass transfer boundary layer of water
has much higher resistance than the gaseous layer for sparingly soluble $Hg^0$, the overall mass transfer coefficient
is limited by water transfer velocity (Eq. 5) (Kim and Fitzgerald, 1986). Surface wind speed is an important
driving force enhancing the mass transfer coefficient in water (Qureshi et al., 2011b), $D_{Hg^0}$ has been
experimentally determined as a function of temperature ($T$, Kelvin) for freshwater ($D_{Hg^0}^{fresh} = 0.0335 e^{-18.63/RT}$, $R$
represents gas constant) and seawater ($D_{Hg^0}^{sea} = 0.0011 e^{-11.06/RT}$) (Kuss, 2014).

Processes controlled the concentration of DGM in surface water directly regulated air-water $Hg^0$ flux.

Photochemically induced $Hg^{II}$ reduction is the predominant pathway of DGM formation in surface water (Amyot
et al., 1994; Amyot et al., 1997a; Amyot et al., 1997b; Costa and Liss, 1999; Lalonde et al., 2001; Zhang and
Lindberg, 2001; Feng et al., 2004). Zhang (2006) summarized Hg photochemical redox chemical process. Eq. (6)
resembling a simplified scheme of gross photo-reactions governing the DGM pool in surface waters (O'Driscoll
et al., 2006; O'Driscoll et al., 2008; Qureshi et al., 2010):

$Hg_{reducible} + photoreductants \rightleftharpoons DGM + photooxidants$ \qquad (4)

$Fe^{III}$ has been reported to enhance sunlit photo-reduction in natural water (Lin and Pehkonen, 1997; Zhang and



Lindberg, 2001). Complexes of $Fe^{III}$-natural organic ligands was hypothesized to undergo photolysis to form
reactive intermediates (e.g., organic free radicals) capable of reducing $Hg^{II}$. Dissolved organic matter (DOM)
serving as electron donor and complexation agent in the natural water is the most important precursor for
formation of photo-reductants (Ravichandran, 2004; Vost et al., 2011; Zhang et al., 2011). Similarly, irradiation
derived photo-oxidant may oxidize DGM simultaneously and reduce Hg evasion from water. Reactive radicals
(e.g., $\cdot O_2$, $\cdot OH$) produced through DOM, $NO_3^-$ photolysis have been identified as possible oxidants (Lin and
Pehkonen, 1997; Zhang and Lindberg, 2001; Zhang et al., 2012b). In addition, $Cl^-$ was reported to enhance
photo-oxidation by stabilizing the oxidative products ($HgCl_n^{2-n}$) and facilitating oxidation via formation of
highly oxidizing ligand ($Cl_2^{\cdot-}$) (Yamamoto, 1996; Lalonde et al., 2001; Sun et al., 2014). Secondary radicals (e.g.,
$CO_3^{\cdot-}$) can sometimes act as a photo-oxidant (He et al., 2014). Field studies also observed DGM and $Hg^0$ flux
peaks in the nighttime, suggesting the importance of dark reduction (O'Driscoll et al., 2003; Zhang et al., 2006b;
Fu et al., 2013b). Dark abiotic redox transformation is the most important pathway (Fig. 5). Although dark
abiotic reduction takes place mainly in the anoxic environment (Gu et al., 2011; Zheng et al., 2012), it also occurs
in the oxic condition at a lower reaction rate (Allard and Arsenie, 1991). Natural organic matter shows reducing,
oxidizing, and complexing properties with Hg in the anoxic environment due to its diversity functional groups
(e.g., thiols group, quinones and non-quinoid structures, carboxyl group) (Gu et al., 2011; Zheng et al., 2012;
Zheng et al., 2013). Although aqueous liquid Hg droplet can be rapidly oxidized in oxygenated chloric water,
DGM is unable to be oxidized under such conditions (Amyot et al., 2005).
Biological redox transformation is another important DGM cycling pathway. Ariya et al. (2015) reviewed
the biological processes in Hg redox transformation, which contains phototrophic and chemotrophic Hg redox
processes. Aquatic algae, cyanobacteria, and diatoms involved phototrophic Hg reduction was positively
correlated with photosynthetic activities, which is likely a bio-detoxification process (Ben-Bassat and Mayer,
1975; Kuss et al., 2015). In addition, photo-reactivation of DOM and $Fe^{III}$ facilitates $Hg^{II}$ reduction through algae
(Deng et al., 2009). Kuss et al. (2015) reported that cyanobacteria-light synergetic and photochemical
transformation equally contributed to ~30% DGM production in Baltic Sea, while low-light production
contributed ~40%, highlighting the importance of biotic reduction. Two pathways have been identified for $Hg^{II}$



reduction by bacteria. The first is reduction by Hg-resistant microorganisms where $Hg^{II}$ is reduced in cell's
cytoplasm by mercuric reductase and transported out as $Hg^0$ (Barkay et al., 2003); the other is $Hg^{II}$ reduced by
Hg-sensitive dissimilatory metal-reducing bacteria utilizing iron and/or manganese as terminal electron acceptor
during respiration (Wiatrowski et al., 2006). Intracellular oxidation was supposed to be mediated by oxidase
(Siciliano et al., 2002), while extracellular thiol functional groups on cell membrane also shows capability to
oxidize $Hg^0$ under anoxic environment (Colombo et al., 2013; Hu et al., 2013). A review of genetic-based
microbial Hg redox transformation can be found in Lin et al. (2011).

**3.4 Air-snow Hg exchange**

Schroeder et al. (1998) reported episodes of unexpected low $Hg^0$ concentrations in the Arctic air during

spring time, so-called atmospheric mercury depletion events (AMDEs), through an arrays of photochemically
initiated oxidation by halogens (Lindberg et al., 2002a; Sommar et al., 2007; Moore et al., 2014). The
phenomena was finally confirmed widespread in the coastal Polar Regions. During AMDEs, a large amount of
surface layer $Hg^0$ is oxidized and deposited in snowpack via GOM and PBM dry deposition (Steffen et al., 2008).
The deposited Hg onto snow can be rapidly re-volatilized back to the atmosphere via photochemical $Hg^{II}$
reduction on snow or in melted snow (Dommergue et al., 2003; Faïn et al., 2007; Kirk et al., 2006).
Photo-reduction is the predominant pathway for Hg reemission from snow as inferred by Hg isotope
fractionation signatures (Sherman et al., 2010). The reduction rate was found to be linearly correlated with UV
intensity (Lalonde et al., 2002; Mann et al., 2015b), while $Cl^-$ showed an inhibiting effect on the photo-reduction
(Section 3.3) (Steffen et al., 2013). Oxidation and reemission of $Hg^0$ occurred simultaneously with the presence
of oxidants (e.g., $\cdot OH$, Cl, and Br) formed through photolysis (Poulain et al., 2004). Nighttime elevated GEM in
snow air was observed at Station Nord, Greenland, likely a result from dark formation of reducing radicals (e.g.,
$HO_2\cdot$) (Ferrari et al., 2004). Temperature is another factor enhancing Hg emission from snow by changing the
solid and liquid water ratio (Mann et al., 2015a). $Hg^0$ flux from snow surface in the temperate regions has rarely
been investigated (Faïn et al., 2007). Field data collected in Ontario and Northern New York confirmed that
photo-reduction is the predominant pathway in enhancing $Hg^0$ emission (Lalonde et al., 2003; Maxwell et al.,
2013). A positive correlation between $Hg^0$ fluxes and temperature has also been found (Maxwell et al., 2013).





$Hg^0$ flux over snow cover under forest canopy was found to be smaller compared to those found in open field,
possibly caused by lower light under canopy (Poulain et al., 2007).

**4. Global observation of atmosphere-biosphere Hg exchange**
**4.1 Data sources, extraction and processing**
A comprehensive database of global observation of $Hg^0$ flux over terrestrial and oceanic surfaces is
compiled from the field observed data reported in peer-reviewed literatures; and the fluxes over water surfaces
calculated using two-film gas exchange model based on *in-situ* measured DGM are also included. For those
studies that measured TGM ($Hg^0$ + GOM) got flux calculation, the measured flux is regarded as $Hg^0$ flux
because of the small fraction of GOM in TGM measurement (GOM/GEM < 2% in general, Gustin and Jaffe,
2010, Sprovieri et al., 2010; Fu et al., 2015), therefore $Hg^0$ and TGM are not discriminable for Hg vapor analyzer
during a typical concentration measurement period (5 mins sampling, 1.0-1.5 L $min^{-1}$) in flux sampling. As
complete time-series flux datasets are not available in literature, each data point included in the database
corresponds to the arithmetic mean of the flux observed during each campaign, with the campaign period lasting
up to one year. For those studies periodically (e.g. weekly) measured seasonal flux at the same site, the average
fluxes of all campaigns was used. A summary of $Hg^0$ flux data documented in a total of 172 peer reviewed
articles are presented in Table 1, which were obtained using DFCs (85.6%), MM (7.9%), $Hg^0/^{222}Rn$ flux ratio
(0.3%), and enriched isotope tracers (0.1%), or estimated using two-film gas exchange model (6.1%). Based on
the landscapes characteristics and surface Hg contents, the flux datasets are assigned into 11 categories.
Classification of background soils (e.g., open field bare soil and forest ground soils with little perturbation by
human activities) follows the corresponding literature definition. Soil Hg content of $\leq 0.3$ $\mu g$ $g^{-1}$ was applied as
the threshold for background soil in case no classification was assigned in the original article. Hg contaminated
sites are divided into natural enriched and anthropogenic contaminated sites based on the Hg sources. The
remaining flux data were categorized into 9 classes according to the land uses and ecosystem types (Table 1). It is
important to recognize that the $Hg^0$ fluxes represent the experimental and modeling results using diverse
methodologies with campaign periods of different durations. Given the reasonably large flux sample sizes, the
flux statistics (e.g., mean, median) from multiple studies for different landscapes are compared. It should be





noted that flux reported in laboratory controlled and field manipulated experimental studies utilizing
treated/untreated substrates are *not* included in the database. Instead, the implications of those studies are
discussed in terms of the environmental effects of $Hg^0$ exchange mechanisms (cf. Section 3).

**4.2 Global database of earth surfaces-atmosphere $Hg^0$ flux**

Table 1 summarizes the statistics of $Hg^0$ fluxes measured to date. The site characteristics where $Hg^0$ flux
measurements were performed are highly diverse. Most studies were devoted to flux investigation over natural
Hg-enriched sites (~38.2%) and background surfaces (~18.4%). Direct field measurements over terrestrial
surfaces accounts for 94.1% (n=811) of the data, only 5.9% (n=51) of the data represents oceanic fluxes. In terms
of substrate Hg contents, measurements at contaminated sites (natural Hg-enriched and anthropogenic polluted)
consisted of 44.9% of the datasets, motivated by extensive emission at these sites caused by local and regional
atmospheric pollution. For unpolluted terrestrial surfaces, most measurements were carried out over background
soils (37%, n=159), while only a few studies directed to the forest foliage and above canopy flux (n=8). DFC
methods are suitable for bare soil and low vegetated surface, covering 97% of the data over background soils.
The remaining datasets are observations of ecosystem flux using MM methods, which require relatively more
complex instrumentation and experimental efforts in the field (Gustin, 2011; Aubinet et al., 2012; Sommar et al.,
2013a).
Fig. 6 shows the box and whisker plots of $Hg^0$ fluxes. As seen, the categorized data exhibit substantial data
variability and positive skewness. Many campaigns focus only on daytime flux (cf. Section 4.3.2) and therefore
the median of mean flux in each category is a more appropriate statistics for comparison. The medians of $Hg^0$
fluxes for the 11 site categories follow the order: grasslands < forest foliage & canopy level < background soils <
wetlands < seawater < snow < freshwater < urban settings < agricultural fields < anthropogenically contaminated
surfaces < natural Hg-enriched surfaces (Table 1). A clear increase in flux from background to contaminated
sites suggests the strong influence of substrate Hg contents on $Hg^0$ flux. Median fluxes from contaminated sites
are two orders of magnitude greater than those over other surfaces; such source strength significantly enhances
local and regional atmospheric Hg concentration. Fluxes over vegetative surfaces (grasslands, forest foliage and
canopy level), mixed vegetated waters (wetlands) are lower than those over background soils and open water



(freshwater and seawater), supporting that vegetation reduces Hg emission by masking ground floor evasion
and/or plant uptake. The fluxes at human perturbed urban settings and over agricultural fields were higher than
the flux over undisturbed earth surfaces, likely a result of reemission of legacy Hg deposition. Most surfaces
showed net $Hg^0$ emission; approximately 25% of measurements over vegetated surfaces showed net Hg
deposition (Fig. 6).
Results of frequency analysis of the mean $Hg^0$ fluxes for each land cover are presented in Fig. 7. While the
mean $Hg^0$ flux from background soils have a large range (-51.7 – 33.3 ng m$^{-2}$ h$^{-1}$), ~90% of the flux data ranges
from -5 to 10 ng m$^{-2}$ h$^{-1}$. Similar patterns are also evident for freshwater, oceans, grasslands, and wetlands. The
occasional high emission and deposition fluxes are mainly due to short sampling duration (e.g. mid-day flux) or
extreme atmospheric $Hg^0$ concentration events caused by local/regional sources. Comparatively, fluxes over
agricultural fields and in urban settings show a much larger range and a lower kurtosis. Strong $Hg^0$ evasion was
observed at contaminated sites (>97% of total observations showed evasion), although extremely high
deposition also occurred in the presence of high ambient $Hg^0$ and atmospheric subsidence (Bash and Miller, 2007;
Zhu et al., 2013c). Most measurements over snow (87%) show evasions; these studies were carried out in the
Polar Regions and focused on Hg reemission from snow after AMDEs. The distribution of $Hg^0$ fluxes of
air-foliage and canopy level exchange showed that half of the measurements (n=4) gave a net emission, while the
mean flux is not significantly different from zero ($p$=0.24, ANOVA).

**4.3 Spatial distribution and temporal variation of global $Hg^0$ flux data**
*4.3.1 Spatial distribution*
Fig. 8 shows the box and whisker plots of $Hg^0$ flux from four relatively homogenized surfaces (background
soils, agricultural fields, grasslands, and freshwater) observed in different regions. Worldwide flux measurement
was unevenly distributed, most studies were conducted in North American and East Asia, which limits global
representativeness. $Hg^0$ flux observed in East Asia is consistently higher compared to those measured in Europe,
North and South America, Australia, and South Africa ($p < 0.05$, ANOVA, except freshwater). This can be
explained by the greater anthropogenic emission and re-emission of Hg deposition (Selin et al., 2007; Selin et al.,
2008; Lin et al., 2010b; Smith-Downey et al., 2010). The flux over freshwater in Europe is somewhat higher than



those measured in East Asia (6.5 vs. 4.6 ng m$^{-2}$ h$^{-1}$, $p$=0.40, ANOVA). These data were obtained mostly prior to
2002 (n=9) or during summer time and daytime (n=8) (Schroeder et al., 1989; Xiao et al., 1991; Lindberg et al.,
1995b; Gårdfeldt et al., 2001; Feng et al., 2002), which could have yielded higher fluxes.

*4.3.2 Diurnal and seasonal patterns*
Fig. 9 displays the general diel variation of Hg$^0$ flux measured by DFC and DFB methods. Fluxes were
typically higher during daytime and lower at nighttime from soil, mine, water and snow surfaces, where Hg$^0$ can
be formed through photo-reduction. As discussed in Section 3, the observed diel variations are in agreement with
results from laboratory controlled studies: higher irradiance and temperature promoted Hg$^0$ reduction and
evasion at daytime, which formed a "dome-shaped" diel flux pattern from most earth surfaces (e.g., soils, mine,
water, and snow). On the contrary, greater deposition during daytime and evasion/near-zero-flux at nighttime
have been frequently observed from foliage, possibly facilitated by the uptake through stomata that exhibit
higher stomatal conductivity during daytime.
Seasonally, higher evasion flux occurs in warm season and smaller exchange is observed in cold season. For
example, seasonal data from Choi and Holsen (2009a) showed a higher evasion from forest floor soil in
Adirondack Mountain (New York, USA) in summer (1.46 ng m$^{-2}$ h$^{-1}$) shifted to insignificant exchange in winter
(0.19 ng m$^{-2}$ h$^{-1}$). Similar trends were also found in agricultural soils, freshwater, and mine surfaces (Fu et al.,
2010a; Eckley et al., 2011b; Zhu et al., 2011). Observed diurnal and seasonal patterns may also be influenced by
vegetative surface changes and meteorological characteristics. For example, Sommar et al. (2015) reported
seasonal flux observation over a wheat-corn rotation cropland using REA measurement, an unexpected low flux
was observed in summer during corn growing stage (median: -6.1 ng m$^{-2}$ h$^{-1}$) due to the uptake by corn leaf (leaf
area index 2.7-3.6), which is similar to the flux (-6.7 ng m$^{-2}$ h$^{-1}$) observed in winter and much lower than the
wheat canopy flux (13.4 ng m$^{-2}$ h$^{-1}$) in early spring. The limited availability of seasonal data in peer-reviewed
literature does not allow a thorough assessment of seasonal characteristics of different terrestrial surfaces. It is
important to recognize that the modified landscapes and vegetative biomass growing cycle caused by seasonal
changes (e.g., change of LAI in deciduous forest, growing season of forest ecosystem, etc.) may significantly
modify the flux characteristics. More data, especially measurements using consistent quantification techniques





over a longer campaign period (e.g., 1 year or longer), are needed for addressing the seasonal variability of $Hg^0$
exchange flux and better estimating the annual exchange from vegetative surfaces. To accomplish such
measurements, automation of flux quantification apparatus is also required.

**4.4 Source and sink characteristics of natural surfaces in the context of global Hg budget**
*4.4.1 Background soils and water are important diffuse sources of $Hg^0$*
Although $Hg^0$ flux observed over background soil (1.3 ng $m^{-2}$ $h^{-1}$) and unpolluted water bodies (2.8 and 2.5
ng $m^{-2}$ $h^{-1}$ for fresh and seawater) may appear mild (Table 1), the annual emission from these two types surfaces
accounted for 64% of total atmospheric Hg emission because of their large areal coverage globally (Pirrone et al.,
2010). For example, it has been estimated that bare soil releases ~550 Mg $yr^{-1}$ (Selin et al., 2008; Pirrone et al.,
2010) and surface ocean releases 2000-2900 Mg $yr^{-1}$ of $Hg^0$ globally (Fig. 1) (Mason et al., 2012; AMAP/UNEP,
2013). Constraining the uncertainties on $Hg^0$ emission from these diffuse sources will greatly improve the
accuracy of global Hg budget. Global $Hg^0$ evasion from soil is mainly based on empirical relationship between
flux, temperature and irradiation (cf., Section 5), which needs mechanistic refinement. Air-seawater exchange
estimated by global models is subject to the uncertainty in (1) mechanisms of aqueous redox transformation and
the associated kinetic parameters, and (2) $Hg^0$ mass transfer rates as determined by surface fiction velocity
(Qureshi et al., 2011a). Kinetic parameters of these processes largely rely on limited field data without
experimental verification (AMAP/UNEP, 2013) and require further investigation. Parameterization of $Hg^0$ flux
using field data and redox transformation rate constants in soil and water are critical to reduce the uncertainty in
future studies.

*4.4.2 Contaminated surfaces are intensive local $Hg^0$ sources*
Hg evasion from contaminated surfaces (Fig. 6 and Fig. 7) has been recognized as an important input
contributing to regional atmospheric Hg budget (Ferrara et al., 1998b; Kotnik et al., 2005). $Hg^0$ flux from
contaminated point sources have been extensively investigated by using LIDAR technique, which is by far the
most effective experimental approach to spatially resolve the $Hg^0$ air-surface exchange at contaminated sites.
Ferrara et al. (1998a) measured the spatial distribution TGM concentration and TGM flux from the world's





largest Hg mine, Almadén Hg mine, in Spain. TGM concentration and flux were estimated to be $0.1 - 5\ \mu g\ m^{-3}$,
$600 - 1200\ g\ h^{-1}$ in fall, 1993, above the village of Almadén. Several attempts have been made to quantitatively
estimate atmospheric Hg input in mining areas. Gustin et al. (2003) and Wang et al., (2005) applied a log-linear
correlation between the flux and substrate Hg contents and solar irradiance. Eckley et al. (2011b) computed
annual Hg emission from two active gold mines (up to $109\ kg\ year^{-1}$) using flux measurement of flux and
statistically derived the empirical relationship between flux and meteorological variables based on Geographical
Information System (GIS) data. Similarly, Kocman and Horvat (2011) obtained ~$51\ kg\ year^{-1}$ emission from
Idrijca River catchment, a former Hg mine, using field measurement and GIS data. In total, annual Hg emission
from global contaminated surfaces was estimated to be ~82 Mg via modeling of flux from more than 3000 Hg
contaminated sites comprising Hg mining, non-ferrous metal production, precious metal processing, and various
polluted industrial sites (Kocman et al., 2013), which is emitted from a very limited surface areas thus can pose a
strong environmental impact to the local area surrounding the contaminated sites.

*4.4.3 Areas impacted by human activities exhibit elevated $Hg^{0}$ reemission*
The median evasion flux over human urban settings and agricultural fields is 5-10 times higher than the
value over background soils (Table 1). Direct anthropogenic Hg input and atmospheric Hg deposition explain the
enhanced reemission. Natural surfaces nearby the anthropogenic point sources (e.g. power plant, Pb-Zn smelter,
chlor-alkali plant) generally showed higher soil Hg content due to atmospheric Hg deposition (Lodenius and
Tulisalo, 1984; Li et al., 2011; Zheng et al., 2011; Guédron et al., 2013). A fraction of these deposited Hg can be
swiftly reemitted back to the atmosphere (Fu et al., 2012a; Eckley et al., 2015). Newly deposited Hg to soil,
aquatic system and snow pack in the Polar Regions can also be readily converted to $Hg^{0}$ and reemitted (Amyot et
al., 2004; Poulain et al., 2004; Ericksen et al., 2005). Eckley et al. (2015) observed soil $Hg^{0}$ flux near a large
base-metal smelter (Flin Flon, Manitoba, Canada) and reported a net deposition during operation ($-3.8\ ng\ m^{-2}\ h^{-1}$)
and elevated emission ($108\ ng\ m^{-2}\ h^{-1}$) after operation ceased. To date, the source and sink characteristics of
surfaces impacted by human activities have not been adequately investigated. Future investigation should be
coordinated toward spatially resolving the $Hg^{0}$ exchange over human impacted surfaces for better quantifying
the emission budget of legacy Hg.






*4.4.4 Flux over vegetated surfaces likely a sink but large uncertainties remains*
Data of $Hg^0$ flux over foliage and forest canopy showed a small net emission (median: 0.7 ng $m^{-2}$ $h^{-1}$) with
substantial variability (Figs. 6 and 7). There have been conflicting reports regarding the role of forest ecosystems
as Hg source or sink at global scale (Lindqvist et al., 1991; Lindberg et al., 1998; Frescholtz and Gustin, 2004;
Fay and Gustin, 2007a; Fay and Gustin, 2007b; Hartman et al., 2009; Cui et al., 2014). Laboratory studies
suggested that plant is a net sink atmospheric Hg through leaf assimilation (Millhollen et al., 2006a;
Stamenkovic and Gustin, 2009; Rutter et al., 2011b; Cui et al., 2014). Using Hg concentration in plant tissues and
net primary productivity as a proxy for atmospheric Hg deposition, Obrist (2007) estimated plants remove
~1024.2 Mg $yr^{-1}$ Hg globally (foliage contributed 237.6 Mg/yr). Fu et al. (2015b) estimated that global litterfall
contributes to 1232 Mg $yr^{-1}$ of Hg deposition, throughfall contributes to 1338 $yr^{-1}$ of Hg deposition, and forest
floor evades ~381 Mg $yr^{-1}$ of Hg into the atmosphere. Hg content in forest soil is comparatively higher than the
concentration found in bare soil due to the input via litterfall and wet Hg deposition (Blackwell and Driscoll,
2015a, b; Obrist et al., 2011); and ~90% boreal forest soil Hg was believed to be originated from litterfall input
(Jiskra et al., 2015). These studies suggested forest ecosystem is likely a large atmospheric Hg sink, although
these bulk proxy methods are not sufficiently sophisticated to resolve the global Hg mass balances.
Synchronized, long-term observation of air canopy flux and litterfall/throughfall deposition is useful to
understand the source and sink characteristics of forest.

**5. Modeling of air-surface $Hg^0$ exchange flux**
A summary of recent modeling efforts on estimating natural emission was presented in Table 2. For
air-foliage $Hg^0$ exchange, earlier parameterization (S1, Table 2) calculates the flux as a function of the
evapotranspiration rate based on soil-root-stem-foliage transpiration stream. It is assumed that Hg passed
through the soil-root interface and then is transferred into foliage in as complexes with organic ligands (Moreno
et al., 2005a; Moreno et al., 2005b; Wang et al., 2012). However, root uptake is unlikely to occur (Cui et al.,
2014). Hg isotopic signatures between air and foliage (Demers et al., 2013; Yin et al., 2013), and air-foliage flux
measurements (Graydon et al., 2006; Gustin et al., 2008) suggest that: (1) the exchange is bi-directional, and (2)



atmospheric Hg uptake by foliage is the major pathway for Hg accumulation. Therefore, a bidirectional flux
scheme building on the compensation point (S2, Table 2) is perhaps more scientifically sound and
mathematically robust. For air-soil $Hg^0$ exchange, in addition to the bidirectional resistance scheme (S3),
statistical relationships have been developed based on measured flux and observed environmental factors such as
air/soil temperature, solar radiation, soil moisture and soil Hg content (S1-S2, Table 2), which tends to be
site-specific and oversimplifies the influence of environmental factors (Wang et al., 2014). For air-water flux
simulation, the two-film diffusion model is widely used by incorporating surface storage and aqueous Hg redox
chemistry (Bash et al., 2007; Strode et al., 2007). Bash (2010) suggested a pseudo-first kinetic water photo-redox
scheme in CMAQ simulation with bidirectional Hg exchange. Strode et al. (2007) parameterized the reduction
rate as the product of local shortwave solar radiation, net primary productivity, and a scaling parameter in
GEOS-Chem. Soerensen et al. (2010) updated the surface ocean redox reactions in GEOS-Chem, and added a
term for dark oxidation, and suggested new linear relationships between the total solar radiation, net primary
productivity, and photo-oxidation rate coefficient, photoreduction coefficient, and biotic reduction coefficient.
Using the S1 scheme (Table 2), the range of simulated air-foliage fluxes were 0 to 5 ng $m^{-2}$ $h^{-1}$ in North
America (Bash et al., 2004) and 0 to 80 ng $m^{-2}$ $h^{-1}$ in East Asia (Shetty et al., 2008). Change the modeling
approach to resistance based models with compensation point assumption (S2 scheme), the range was -2.2 to
-0.7 ng $m^{-2}$ $h^{-1}$ (Wang et al., 2014). Zhang et al. (2012a) reported the annual $Hg^0$ uptake by foliage was 5-33 μg
$m^{-2}$ with the S2 scheme, similar to the litterfall Hg flux measured at Mercury Deposition Network Sites. For
air-soil exchange, model-estimated fluxes ranged from 0 to 25 ng $m^{-2}$ $h^{-1}$ using the S1 and S2 scheme (Bash et al.,
2004; Gbor et al., 2006; Shetty et al., 2008; Kikuchi et al., 2013), comparable to the 0-20 ng $m^{-2}$ $h^{-1}$ using the S3
scheme (Wang et al., 2014). For air-water exchange, the model-estimated flux was 1-12 ng $m^{-2}$ $h^{-1}$, consistent
with measured fluxes (Bash et al., 2004; Shetty et al., 2008; Bash, 2010; Wang et al., 2014).
Future development of $Hg^0$ flux model requires mechanistic understanding of air-surface exchange
processes. Presently, bidirectional resistance scheme, the stomatal compensation point is treated as a constant
value (Bash, 2010; Wang et al., 2014) or calculated as following in Wright and Zhang (2015):

$$\chi_{st} = 8.204 \frac{8.9803 \times 10^9}{T} \cdot \Gamma_{st} \cdot e^{-\frac{8353.8}{T}} \tag{5}$$





where T is the temperature of stomata/surface, and $\Gamma_{st}$ is the emission potential of the stomata. $\Gamma_{st}$ is an empirical
input value and suggested in 5-25 ng m$^{-3}$ depending on the specific land use. Battke et al., 2005; Heaton et al.,
2005; and Battke et al., 2008 reported that plants have the ability to reduce the $Hg^{II}$ to $Hg^0$ in foliar cell through
reducing ligands (e.g., NADPH). To propose a more physically robust modeling scheme, the redox processes in
foliage and the role of ligands on Hg uptake need to be better understood. The finding that $Hg^0$ can pass through
the soil-root interface under artificial laboratory conditions (Moreno et al., 2005b) needs to be carefully verified
in the field.
Another area that requires advancement is the determination of $Hg^{II}$ reduction rate (Scholtz et al.,
2003;Bash, 2010;Wang et al., 2014) and the hypothetical parameter $\Gamma_{st}$ (Wright and Zhang, 2015) in soil. It is
well known that $Hg^{II}$ can be reduced by natural organic acids via biotic/abiotic reduction (Zhang and Lindberg,
1999; Zheng et al., 2012). Experimental investigations showed that $O_3$ is important in controlling Hg emissions
from substrates (Engle et al., 2005). However, the kinetic description of these process is fundamentally unknown.
The pseudo-first reduction rate constant of $Hg^{II}$ has been assumed to be in the range of $10^{-11}$ to $10^{-10}$ s$^{-1}$ (Scholtz
et al., 2003; Qureshi et al., 2011a). Under laboratory conditions at 100 W m$^{-2}$ and 32±7 °C, the pseudo-first
reduction rate was estimated to be 2-8×10$^{-10}$ m$^2$ s$^{-1}$ w$^{-1}$ basing on 2 mm soil depth (the maximum depth for light
penetration in soil) (Quinones and Carpi, 2011). Si and Ariya (2015) reported a photo-reduction rate of $Hg^{II}$ in
presence of alkanethiols to be 3-9×10$^{-9}$ m$^2$ s$^{-1}$ w$^{-1}$. Other than these kinetic information, kinetic measurements for
$Hg^{II}$ reduction in the absence of light will enable additional mechanistic parameterization of Hg evasion model
for soil and vegetative surface.

## 6. Conclusions and future perspectives

Understanding in the air-surface exchange of $Hg^0$ has been steadily advancing since mid-1980s. Substantial
amount of data exists, but with large uncertainty and data gaps in Africa, South and Central Asia, Middle East,
South America and Australia. Fundamentally, flux measurement approaches (e.g., MM and DFCs) are different
and individual flux measurement data are not directly comparable. The $Hg^0$ flux data compiled in this study
represent the current state of understanding that requires continuous updates. $Hg^0$ flux in East Asia is statistically
higher than the values observed in other world regions, suggesting reemission of atmospheric deposition or





strong anthropogenic influence. $Hg^0$ exchange over weak diffuse sources (e.g., background soil and water) and
vegetation need better constrains for global analysis of atmospheric Hg budget through extensive on-site
measurement and fundamental mechanical studies (e.g., redox transformation rate constant, mass transfer
diffusivity). Although predominate factors in controlling $Hg^0$ flux have been identified, the effects of those
factors on flux have not been fundamentally and quantitatively determined for different surfaces, which limited
the accuracy of flux modeling. Based on the data synthesis in this study, the following knowledge gaps need to be
addressed:
**(1) Improving temporal resolution and sensitivity of $Hg^0$ flux measurements.** Insufficient temporal
resolution and sensitivity in the detection of ambient Hg has limited our capability in accurately determining the
air-surface exchange of $Hg^0$. Development of high temporal resolution and sensitive sensor for determining $Hg^0$
concentration gradient is of prime importance to improve flux data quality and to reduce uncertainty in the global
assessment of Hg budget. Such advancement will also open up new opportunities to explore fundamental
exchange mechanism in response to the changes in environmental factors.
**(2) Standardization of $Hg^0$ flux measurement techniques and establish data comparison strategy.** $Hg^0$ flux
measurement uncertainties from using different techniques remains large, standardized method is useful to
compare flux obtained from various techniques. Fundamental study is needed to compare current $Hg^0$ flux
quantification methods, synchronized measuring flux from various methods using varied operation parameters is
suggested to build potential empirical data comparison strategy and correction methods, this will largely reduce
the gross uncertainty in the Hg budget estimation and greatly improve comparability of flux data reported by
different research groups.
**(3) Fundamental investigation on the environmental processes driving Hg exchange.** Although flux
response to environmental parameters (e.g., irradiance, precipitation, temperature rising) are qualitatively
defined in statistical sense, the processes driving $Hg^0$ exchange need to be understood fundamentally. Recent
advancement on isotopic tracing techniques (e.g., enriched Hg isotope tracers and stable Hg isotopic
fractionation data) may offer mechanistic insights and new data should be incorporated into new modeling
analysis.
**(4) Long-term measurement of $Hg^0$ flux at representative sites.** There is a substantial data gap in the current





Hg$^0$ flux database in terms of geographical coverage and land use type. Forest is most likely an overlooked sink
for atmospheric Hg$^0$, however, few field campaigns have been conducted at forest sites. In addition, current flux
database are mainly from short-term campaigns. It is presently unclear how global changes (e.g. climate change,
global anthropogenic Hg reduction) will force Hg$^0$ flux changes over different surfaces. There is presently no
network of flux measurements at global monitoring sites. Continuous observation of flux is also useful for
providing better database for scale-up estimation.
**(5) Development and improvement of air-surface exchange models for Hg.** The present state of development
of air-surface exchange model does not allow appropriate process analysis due to a lack of fundamental
understanding in the chemical and mass transfer processes of evasion and deposition. Existing air-surface Hg$^0$
flux schemes incorporate over-simplified chemical schemes with not-yet verified kinetic parameters. In addition,
the interactions between Hg$^{II}$ and organic matters in the natural environment, as well as the interfacial transfer of
different Hg species over various surfaces, have significant knowledge gaps. Studies address these gaps are
critically needed and will benefits not only the measurement approaches but also the model parameterization in
estimating the global air-surface exchange of Hg.

**Acknowledgements**
This research was supported by 973 Program (2013CB430002) and the National Science Foundation of China

750    (41503122).




**Table 1:** A statistical summary of field *in situ* observed Hg$^0$ flux reported in the literatures.

| Landscapes | Hg$^0$ flux (ng m$^{-2}$ h$^{-1}$) | | | | N | References[b] |
|---|---|---|---|---|---|---|
| | Mean | Median | Min (*min*)[a] | Max (*max*)[a] | | |
| Background soil | 2.1 | 1.3 | -51.7 (*-51.7*) | 33.3 (*97.8*) | 159 | *(1)* |
| Urban settings | 16.4 | 6.2 | 0.2 (*-318*) | 129.5 (*437*) | 29 | *(2)* |
| Agricultural fields | 25.1 | 15.3 | -4.1 (*-1051*) | 183 (*1071*) | 59 | *(3)* |
| Forest foliage & canopy level | 6.3 | 0.7 | -9.6 (*-4111*) | 37.0 (*1000*) | 8 | *(4)* |
| Grasslands | 5.5 | 0.4 | -18.7 (*-989.6*) | 41.5 (*870*) | 38 | *(5)* |
| Wetlands | 12.5 | 1.4 | -0.3 (*-375*) | 85 (*677*) | 23 | *(6)* |
| Freshwater | 4.0 | 2.8 | -0.3 (*-18.2*) | 74.0 (*88.9*) | 93 | *(7)* |
| Sea water | 5.9 | 2.5 | 0.1 (*-2.7*) | 40.5 (*46.0*) | 51 | *(8)* |
| Snow | 5.7 | 2.7 | -10.8 (*-2160*) | 40 (*720*) | 15 | *(9)* |
| Natural enriched surfaces | 5618 | 226 | -5493 (*-9434*) | 239200 (*420000*) | 329 | *(10)* |
| Anthropogenically contaminated surfaces | 595 | 184 | -1.4 (*-286.2*) | 13700 (*13700*) | 58 | *(11)* |

Notes: [a]. Min/Max are campaign/site-based average flux, while *(min)/(max)* represent lowest/largest instantaneous flux; [b] References: see **Appendix A**.




**Table 2:** A comparison of natural surface mercury flux models

| | General models | Description | References |
|---|---|---|---|
| **Foliage** | S1:<br><br>$F = EC_s$ | E: transpiration rate (g m$^{-2}$ s$^{-1}$) | Xu et al., 1999; Bash et al., 2004; Shetty et al., 2008; Gbor et al., 2006 |
| | | Cs: Hg$^0$ in soil water (ng g$^{-1}$) | |
| | S2:<br><br>$F_{st/cu} = \dfrac{\chi_{st/cu} - \chi_c}{R_{st/cu}}$ | $\chi_{st/cu}$: stomatal/cuticular compensation point (ng m$^{-3}$) | Zhang et al., 2009;Bash, 2010;Wang et al., 2014;Wright and Zhang, 2015 |
| | | $F_{st/cu}$: air-cuticular/stomatal flux (ng m$^{-2}$ s$^{-1}$) | |
| | | $\chi_c$: compensation point at the air-canopy (ng m$^{-3}$) | |
| | | $R_{st/cu}$: resistance between air-cuticular/stomatal (m s$^{-1}$) | |
| **Soil** | S1:<br><br>$\log F = -\dfrac{\alpha}{T} + \beta \log(C) + \gamma R + \varepsilon$ | T: soil temperature (°) | Xu et al., 1999; Bash et al., 2004; Gbor et al., 2006; Shetty et al., 2008; Selin et al., 2008 |
| | | C: soil Hg concentration (ng g$^{-1}$) | |
| | | R: solar radiation (W m$^{-2}$) | |
| | S2:<br><br>$\dfrac{F}{C} = \alpha T + \beta R + \delta\Theta + \delta TR + \cdots$ | T: soil temperature (°) | Lin et al., 2010a; Kikuchi et al., 2013 |
| | | C: soil Hg concentration (ng g$^{-1}$) | |
| | | R: solar radiation (W m$^{-2}$) | |
| | | Θ: soil moisture (%) | |
| | S3:<br><br>$F = \dfrac{\chi_s - \chi_c}{R_g + R_{ac}}$ | $\chi_s$: soil compensation point (ng m$^{-3}$) | Zhang et al., 2009; Bash, 2010; Wang et al., 2014; Wright and Zhang, 2015 |
| | | $\chi_c$: compensation point at the air-soil (ng m$^{-3}$) | |
| | | $R_g$: resistance between air-soil (m s$^{-1}$) | |
| | | $R_{ac}$: In-canopy aerodynamic resistance (m s$^{-1}$) | |
| **Water** | $F = \dfrac{\chi_w - \chi_c}{R_w + R_a}$ | $\chi_w$: water compensation point (ng m$^{-3}$) | Xu et al., 1999; Bash et al., 2004; Gbor et al., 2006; Shetty et al., 2008; Bash, 2010; Wang et al., 2014 |
| | | $\chi_c$: air Hg$^0$ concentration (ng m$^{-3}$) | |
| | | R$_w$: liquid side resistance (m s$^{-1}$) | |
| | | R$_a$: air side resistance (m s$^{-1}$) | |


**Fig. 1.** The most recent Hg reservoirs and global atmosphere Hg inventory illustrating the exchange flux
between atmosphere and earth surfaces. Adapted from Selin, 2009; Gustin and Jaffe, 2010; Soerensen et al.,
2010; Corbitt et al., 2011; Mason et al., 2012; AMAP/UNEP, 2013.

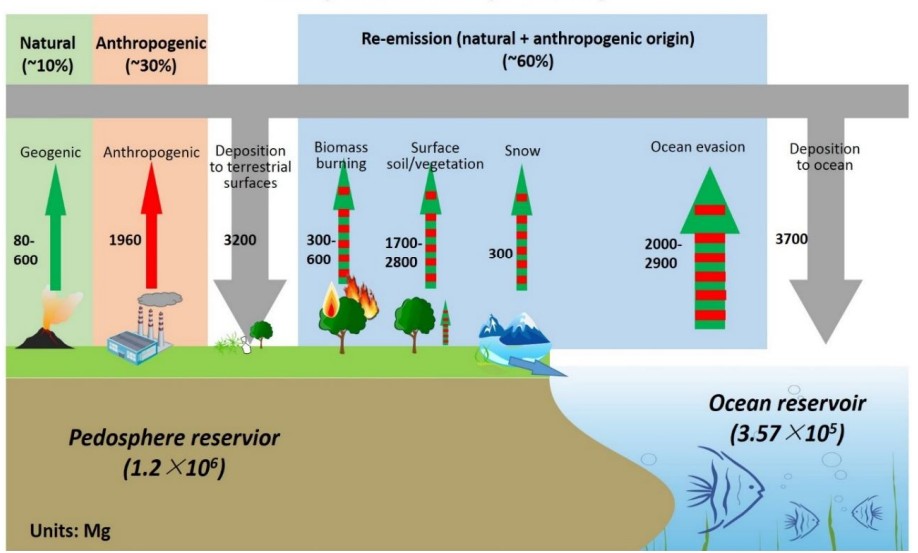






**Fig. 2.** Field collocated DFCs and MM techniques observed fluxes from two intercomparison studies: (a).
Inlaid scatter plot of averaged 4-MM flux vs. averaged 7-DFC flux (TOT: 1.1-24 min) in Hg-enriched
Nevada STORMS site in September, 1997 (Gustin et al., 1999); (b). Diel evolution of Hg$^0$ flux measured
using a 1L polycarbonate-DFC (TOT: 0.2 min) and MBR method at same Nevada STORMS site in October,
1998 (Gustin, 2011); (b). Scatter plot of DFC with traditional/novel designs (TDFC/NDFC) vs. MBR Hg$^0$
flux obtained in Yucheng Intercomparision project (Zhu et al., 2015b).

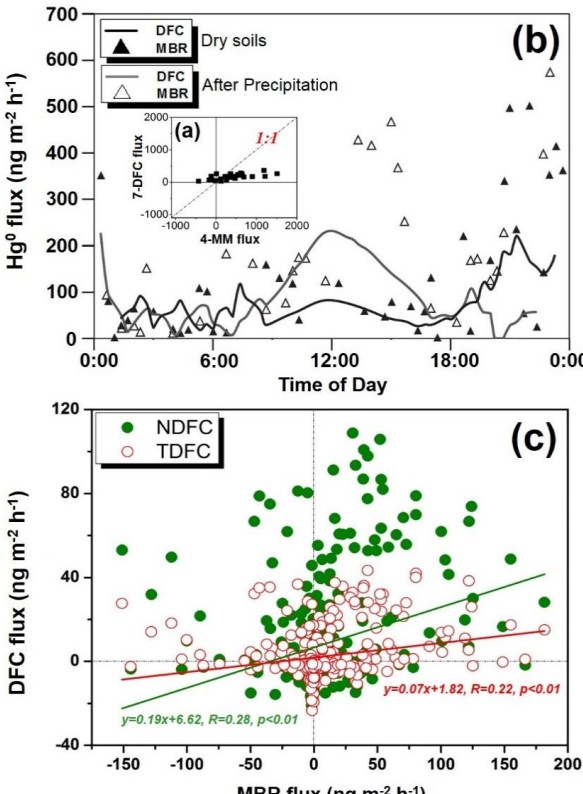







**Fig 3.** Box and whisker plots of Hg$^0$ fluxes measured by MM methods and DFC methods. (Flux data
including measurements from background soils, agricultural fields, grasslands, and wetlands at substrate
total Hg lower than 0.3 µg Hg g$^{-1}$, data source: Table 1. The two box horizontal border lines indicate 25th and
75th percentiles, whiskers represent 10th and 90th percentiles, and outliers (green circles) indicate 5th and
95th percentiles from bottom to top. Red line and black line indicate mean and median flux).

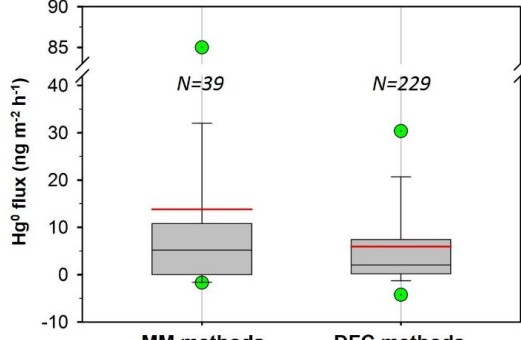




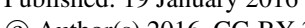



**Fig. 4.** 4-D graphical visualization of the effect of air temperature, soil water content, and solar radiation on
the measured $Hg^0$ flux from soil (Lin et al., 2010a).

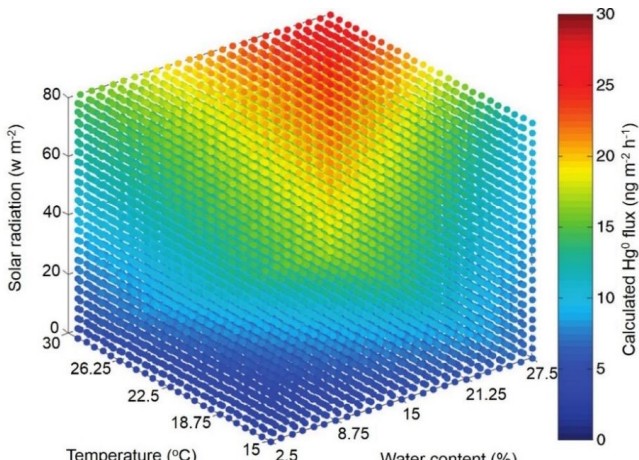






**Fig. 5.** Conceptual view of DGM cycling in water and mass transfer process across the atmosphere-water
interface.

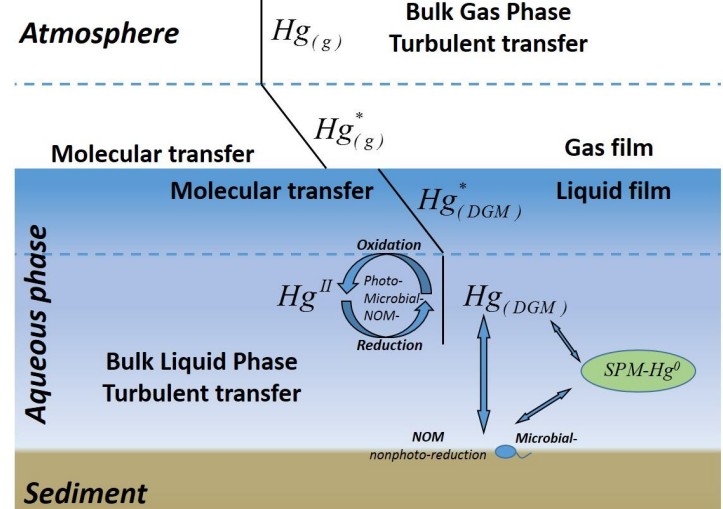






Fig. 6. Box and whisker plots of global field observed Hg$^0$ flux obtained from various landscapes. (Data
Source: Table 1. Red line and black line indicate mean and median flux).

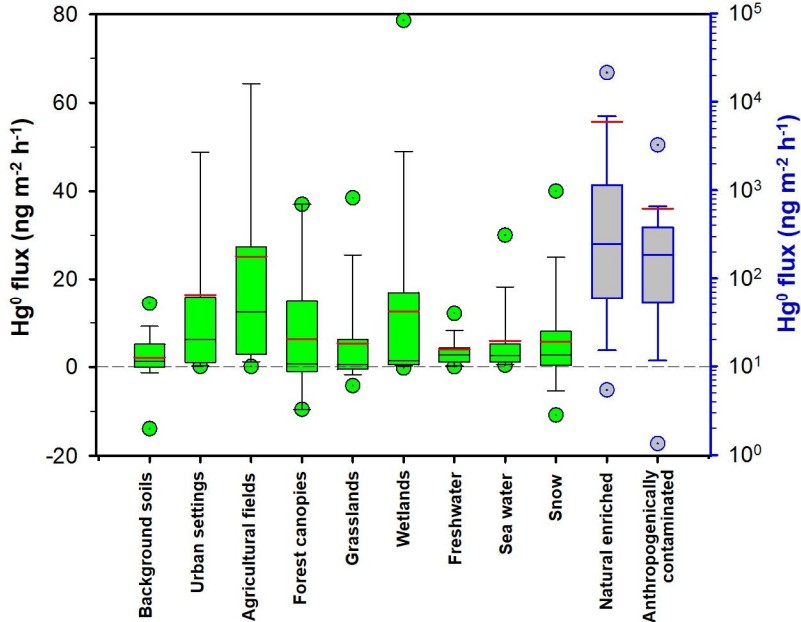








**Fig. 7.** Histograms of Hg$^0$ flux frequency distribution obtained from various earth surfaces. (Data source:
Table 1).

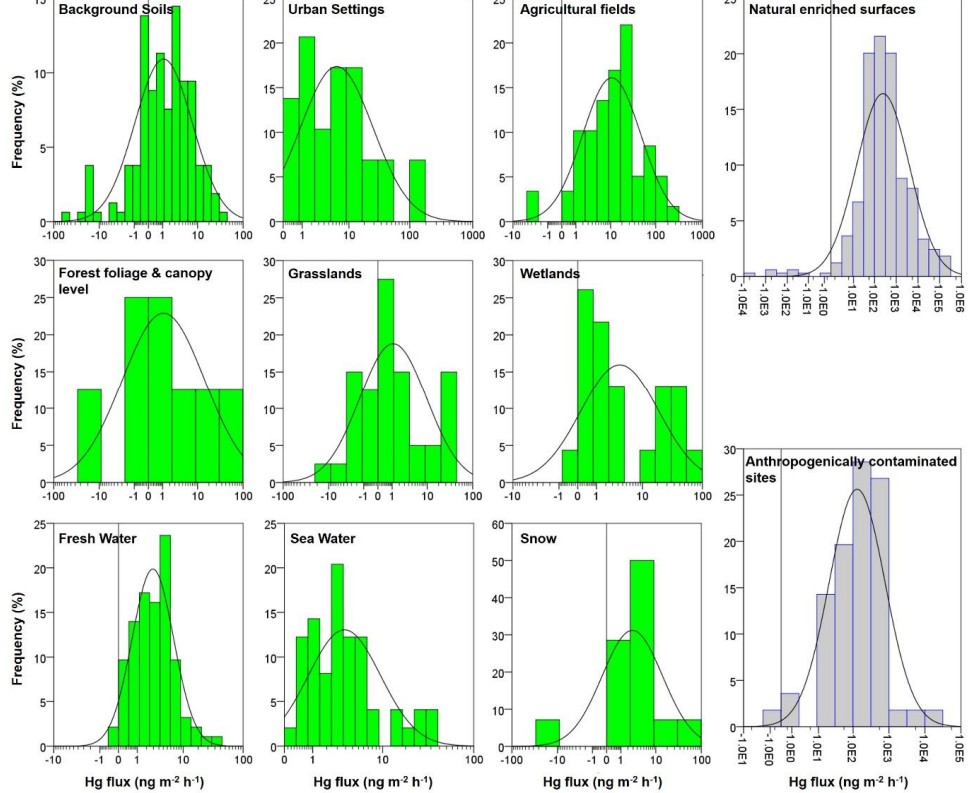








**Fig. 8.** Box and whisker plots of continents segregated Hg$^0$ flux obtained from four homogeneous surfaces
(Background soils, agricultural fields, grasslands, and freshwater. Filled square block and horizontal line in
box indicate mean and median flux).

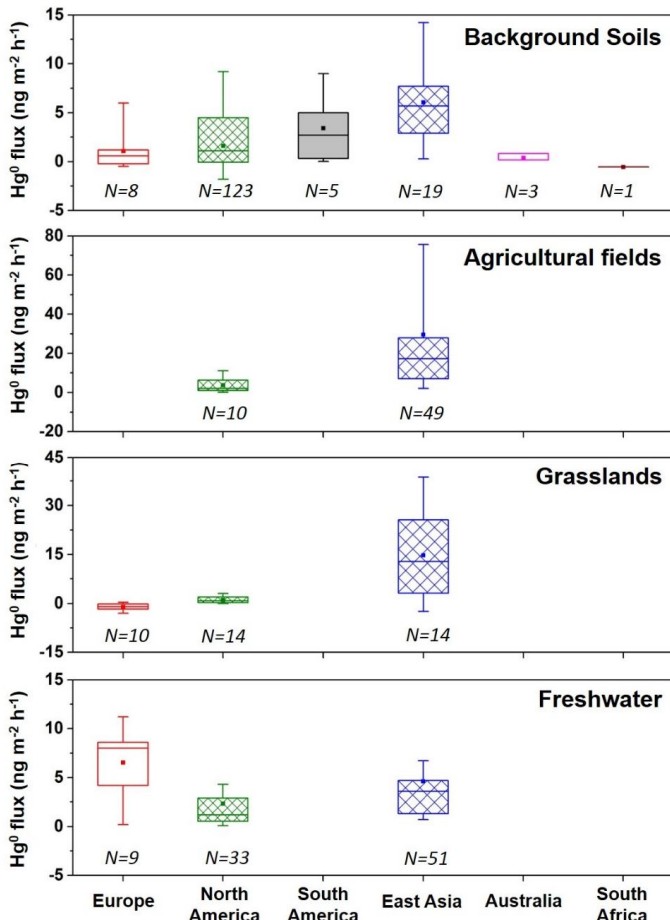








**Fig. 9.** Diurnal patterns of Hg$^0$ flux from various environmental compartments (soil, mine, freshwater, forb
leaf, growing broad leaf, and snow) measured using DFC methods. (Data obtained from soil: Zhu et al.,
2015b; mine: Eckley et al., 2011a; fresh water: O'Driscoll et al., 2003; forb leaf: Stamenkovic et al., 2008;
growing broad leaf: Fu et al., 2015c; and snow: Maxwell et al., 2013)

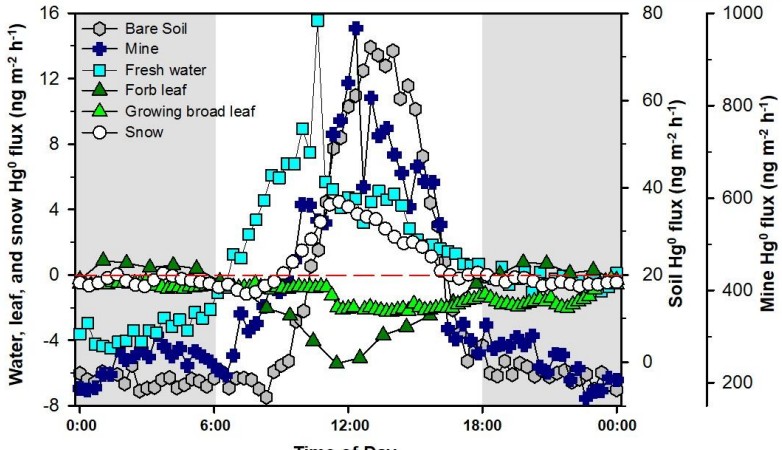








## Appendix A: References for Table 1

*(1)*. Schroeder et al. 1989; Xiao et al., 1991; Kim et al., 1995; Carpi and Lindberg, 1998; Ferrara et al., 1998a; Lindberg et al., 1998; Poissant and Casimir, 1998; Engle et al., 2001; Zhang et al., 2001; Coolbaugh et al., 2002; Hintelmann et al., 2002; Zehner and Gustin, 2002; Fang et al., 2003; Nacht and Gustin, 2004; Poissant et al., 2004b; Edwards et al., 2005; Ericksen et al., 2005; Magarelli and Fostier, 2005; Schroeder et al., 2005; Ericksen et al., 2006; Gustin et al., 2006; Sigler and Lee, 2006; Wang et al., 2006; Fu et al., 2008a; Kuiken et al., 2008a; Kuiken et al., 2008b; Almeida et al., 2009; Choi and Holsen, 2009a; Fu et al., 2012a; Kyllonen et al., 2012; Demers et al., 2013; Edwards and Howard, 2013; Ma et al., 2013; Slemr et al., 2013; Zhu et al., 2013b; Blackwell et al., 2014; Carpi et al., 2014; Du et al., 2014; Fu et al., 2015c;

*(2)*. Kim and Kim, 1999; Fang et al., 2003; Feng et al., 2005; Gabriel et al., 2005; Gabriel et al., 2006; Obrist et al., 2006; Wang et al., 2006; Eckley and Branfireun, 2008; Liu et al., 2014; Osterwalder et al., 2015;

*(3)*. Feng et al., 1997; Carpi and Lindberg, 1998; Cobos et al., 2002; Kim et al., 2002; Kim et al., 2003; Wang et al., 2004; Feng et al., 2005; Schroeder et al., 2005; Ericksen et al., 2006; Xin et al., 2006; Cobbett and Van Heyst, 2007; Fu et al., 2008a; Baya and Van Heyst, 2010; Zhu et al., 2011; Fu et al., 2012a; Sommar et al., 2013b; Zhu et al., 2013a; Zhu et al., 2015b; Sommar et al., 2015;

*(4)*. Lindberg et al., 1998; Graydon et al., 2006; Bash and Miller, 2008; Poissant et al., 2008; Bash and Miller, 2009; Fu et al., 2015c;

*(5)*. Poissant and Casimir, 1998; Schroeder et al., 2005; Ericksen et al., 2006; Obrist et al., 2006; Fu et al., 2008a; Fu et al., 2008b; Fritsche et al., 2008b; Fritsche et al., 2008c; Converse et al., 2010;

*(6)*. Lee et al., 2000; Lindberg and Zhang, 2000; Lindberg and Meyers, 2001; Lindberg et al., 2002b; Wallschläger et al., 2002; Poissant et al., 2004a; Poissant et al., 2004b; Marsik et al., 2005; Schroeder et al., 2005; Zhang et al., 2005; Zhang et al., 2006b; Smith and Reinfelder, 2009; Kyllonen et al., 2012; Fritsche et al., 2014; Osterwalder et al., 2015;

*(7)*. Schroeder et al., 1989; Xiao et al., 1991; Schroeder et al., 1992; Lindberg et al., 1995b; Amyot et al., 1997a; Amyot et al., 1997b; Mason and Sullivan, 1997; Poissant and Casimir, 1998; Boudala et al., 2000; Poissant et al., 2000; Gårdfeldt et al., 2001; Feng et al., 2002; Feng et al., 2003; O'Driscoll et al., 2003; Feng et al., 2004; Hines and Brezonik, 2004; Tseng et al., 2004; Schroeder et al., 2005; Wang et al., 2006; Zhang et al., 2006a; Southworth et al., 2007; O'Driscoll et al., 2007; Feng et al., 2008b; O'Driscoll et al., 2008; Fu et al., 2010a; Fu et al., 2013a; Fu et al., 2013b;

*(8)*. Kim and Fitzgerald, 1986; Mason and Fitzgerald, 1993; Mason et al., 1993; Baeyens and Leermakers, 1998; Mason et al., 1998; Mason et al., 1999; Ferrara and Mazzolai, 1998; Ferrara et al., 2001; Gårdfeldt et al., 2001; Mason et al., 2001; Rolfhus and Fitzgerald, 2001; Wängberg et al., 2001a; Wängberg et al., 2001b; Feng et al., 2002; Conaway et al., 2003; Gårdfeldt et al., 2003; Laurier et al., 2003; Schroeder et al., 2005; St. Louis et al., 2005; Temme et al., 2005; Narukawa et al., 2006; Andersson et al., 2007; Kuss and Schneider, 2007; Sommar et al., 2007; Andersson et al., 2008; Castelle et al., 2009; Fu et al., 2010b; Bouchet et al., 2011; Andersson et al., 2011; Ci et al., 2011a; Ci et al., 2011b; Xu et al., 2012; Fantozzi et al., 2013; Ci et al., 2015; Marumoto and Imai, 2015;

*(9)*. Schroeder et al., 2003; Ferrari et al., 2005; Schroeder et al., 2005; Brooks et al., 2006; Cobbett et al., 2007; Faïn et al., 2007; Sommar et al., 2007; Fritsche et al., 2008c; Steen et al., 2009; Maxwell et al., 2013;

*(10)*. Ferrara et al., 1997; Feng et al., 1997; Ferrara et al., 1998a; Ferrara and Mazzolai, 1998; He et al., 1998; Gustin et al., 1999; Lindberg et al., 1999; Poissant et al., 1999; Wallschläger et al., 1999; Edwards et al., 2001; Engle et al., 2001; Coolbaugh et al., 2002; Engle and Gustin, 2002; Zehner and Gustin, 2002; Gustin et al., 2003; Nacht and Gustin, 2004; Nacht et al., 2004; Edwards et al., 2005; Kotnik et al., 2005; Schroeder et al., 2005; Wang et al., 2005; Engle et al., 2006; García-Sánchez et al., 2006; Wang et al., 2007a; Wang et al., 2007b; Eckley et al., 2011a; Edwards and Howard, 2013; Fantozzi et al., 2013; Dalziel and Tordon, 2014;

*(11)*. Lindberg et al., 1995a; Carpi and Lindberg, 1997; Lindberg and Price, 1999; Kim et al., 2001; Wängberg et al., 2003; Goodrow et al., 2005; Lindberg et al., 2005; Olofsson et al., 2005; Wang et al., 2006; Xin et al., 2006; Nguyen et al., 2008; Rinklebe et al., 2009; Li et al., 2010; Zhu et al., 2013b; Eckley et al., 2015.



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
