# Peer review of "Global observations and modeling of atmosphere-surface exchange of elemental mercury: a critical review"

_Atmospheric Chemistry and Physics, 2015_

## Referee Comment (RC1) · Anonymous Referee #1 · 2 Feb 2016

Overall comments: Overall this is a very well written and fully researched review paper. In addition to simply reviewing existing studies, this paper performs original analysis of the compiled datasets in order to make large scale observations. This paper provides a comprehensive assessment of the current understanding of the atmospheric-surface exchange of Hg and recommend it for publication in Atmospheric Chemistry and Physics.

The only complicating factor with the publication of this paper is that a separate group published a fairly similar paper very recently: Agnan et al., New Constraints on Terrestrial Surface-Atmosphere Fluxes of Gaseous Elemental Mercury Using a Global Database. ES&T 2016 Abstract Despite 30 years of study, gaseous elemental mercury

(Hg(0)) exchange magnitude and controls between terrestrial surfaces and the atmosphere still remain uncertain. We compiled data from 132 studies, including 1290 reported fluxes from more than 200 000 individual measurements, into a database to statistically examine flux magnitudes and controls. We found that fluxes were unevenly distributed, both spatially and temporally, with strong biases toward Hg-enriched sites, daytime and summertime measurements. Fluxes at Hg-enriched sites were positively correlated with substrate concentrations, but this was absent at background sites. Median fluxes over litter- and snow-covered soils were lower than over bare soils, and chamber measurements showed higher emission compared to micrometeorological measurements. Due to low spatial extent, estimated emissions from Hg-enriched areas (217 MgÂůa(-1)) were lower than previous estimates. Globally, areas with enhanced atmospheric Hg(0) levels (particularly East Asia) showed an emerging importance of Hg(0) emissions accounting for half of the total global emissions estimated at 607 MgÂůa(-1), although with a large uncertainty range (-513 to 1353 MgÂůa(-1) [range of 37.5th and 62.5th percentiles]). The largest uncertainties in Hg(0) fluxes stem from forests (-513 to 1353 MgÂůa(-1) [range of 37.5th and 62.5th percentiles]), largely driven by a shortage of whole-ecosystem fluxes and uncertain contributions of leaf-atmosphere exchanges, questioning to what degree ecosystems are net sinks or sources of atmospheric Hg(0).

Given the similarities in objective and scope, I think this paper by Zhu et al, needs to: 1) acknowledge this separate paper in results and discussion and 2) specifically identify how their paper is unique from Agnan et al., 2016, and 3) discuss similarities and differences in the two papers findings.

Specific comments: Line 261: Very interesting result. Glad to see this analysis. A little more information is needed. The samples sizes are 229 and 39, but it is not clear if these numbers represent daily average values, hourly values, etc. A little more discussion about what constitutes a measurement would be helpful. Also, within the <0.3 ug/g cutoff, were there significant differences in the Hg concentrations between

DFC and MM areas? If not, this would help build the case for the analysis.

Line 262: Why was 0.3 ug/g used as a cut-off point.

Line 288: change matters to matter.

Line 285: Two factors that have been shown to affect soil-air Hg fluxes are grain size and soil disturbance. Only a couple of studies have shown this, but may want to consider including these two factors in the discussion if the goal is to be comprehensive as possible.

Line 339. There is a paper by Mazur et al. 2014 in Science of the Total Environment that has a similar focus: the impact of forestry operations on surface-air Hg fluxes.

Line 346: Suggest changing to "more recent" instead of just "recent". This idea has been around for more than a decade now.

Line 352: remove excess Hg0.

Line 422: Need more information to support this statement. Earlier the text focuses on photo-pathways and this is a big jump without sufficient explanation.

Line 472: remove "got flux calculation".

Line 476: in "the" literature

Line 558: This paragraph should also discuss the work of Kuiken et al, 2008 part 1, which shows the opposite trend....lower emission in summer due to drier conditions and lower light from more leaf cover. In the scaling paper, Hartman et al, 2009 comes to the same conclusion.

Line 601. Double check that Gustin et al, 2003 used a multivariate approach using soil Hg, flux and solar radiation. Or did that paper look at these variables separately.

Line 700. Remove "in"

Line 713. This is a great summary of knowledge gaps, glad to see this in the paper.

---

## Referee Comment (RC2) · Anonymous Referee #2 · 9 Feb 2016

Interactive comment on "Global Observations and Modeling of Atmosphere-Surface Exchange of Elementary Mercury – A Critical Review" by W. Zhu et al.

Anonymous Referee #2

General Comments:

This paper is a thorough review of measurement and modeling studies of elemental mercury. The depth and extent of the analyses of available data does indeed make this a critical review rather than just a literature review. The discussion on the advances in the measurement techniques is beneficial. This paper provides a necessary addition to the scientific community's GEM literature and aids in furthering our understanding

of the air-surface exchange of atmospheric mercury. With some minor editing on a technical scale, I recommend the publication of this paper in Atmospheric Chemistry and Physics.

I agree with Reviewer #1's comments regarding the paper by Agnan et al. (2015). The only discussion on their paper was relating to the measurement method. It would be interesting to see a discussion on the findings of the two papers and how they complement each other.

Specific Comments:

Line 108: Is this a possible typo that <1 Hz is considered a higher frequency?

Line 113: The use of "but" in this sentence suggests that the higher detection limit of 0.35 ng m-3 is a negative aspect but could that sensitivity be considered a benefit of this sensor over previous ones?

Lines 185-187: This sentence, while accurate, discusses the lack of the ability of this sensor at background sites. This study however was over Hg-enriched soils and the sensor performed well over Hg-enriched sites. Would it be useful to note this as an advantage to this method considering the high number of sites that are Hg-enriched?

Line 547: Perhaps consider mentioning why the fluxes would be higher in Europe than East Asia prior to 2002 and during summer and/or daytime.

Line 558: There are some studies that suggest the opposite (e.g. Lee et al., 2000; Fristche et al., 2008).

Lines 577; 593; 612; 627: The titles of the subsections in Section 4.4 include statements. Does this possibly change the flow of the paper?

Technical Corrections:

Please see separate file on technical corrections.

---

## Referee Comment (RC3) · Anonymous Referee #2 · 12 Feb 2016

Interactive comment on "Global Observations and Modeling of Atmosphere-Surface Exchange of Elementary Mercury – A Critical Review" by W. Zhu et al.

Anonymous Referee #2

Technical Corrections:

Line 18: Change "air-surfaces flux" to "air-surface fluxes". Line 20: Change "devoting" to "devoted". Line 21: In "the" past three decades. . .. Change "uncertainty remains" to uncertainties remain". Line 24: Change "air-surfaces" to "air-surface". Line 28: "and the" presence. . .. Change "drives" to "drive". Line 29: "the" effects. Line 32: on "the" global. . . Change "flux" to "fluxes". . . "measurement" to "measurements". Line 33:

Change "flux" to "fluxes". Line 36: of "the" evasion flux. Line 40: Change "constrains" to "constraints". . . . "analysis of atmospheric" to "analyses of the atmospheric". Line 44: Change "concerns" to "concern". Line 57: Remove "the". Line 60: Change "velocity" to "velocities". Line 61: Change "hemisphere" to "hemispheric". Line 63: Remove "the". . . Change "cycle" to cycles". Line 69: Change "gradient" to "gradients". Line 71: Change "measurement" to "measurements". . . in "the" 1980s. Line 72: Change "advancement" to "advancements". . . "chamber" to "chambers". Line 73: "the" Hg0/222Rn. . . "the" open-path. . . and "the" Hg0/CO. Line 77: Remove "the" from "in the peer-reviewed". Line 78: Change "literatures" to "literature". Line 93: Change "accounted" to "account". Line 101: Application of "the" appropriate. Line 105: which "relies" on this principle. Line 109: "a" Lumex. Line 110: limit "of" ∼1 ng m-3. Line 111: Change "concentration" to "concentrations". . . Remove "K.". . . recently, "a" high. Line 113: suffers from "the" sensor's. Line 114: technique, "the" laser-induced. Line 116-117: Change "not been yet proved" to "not yet been proven for application in long-term field measurements". Line 120: Change "exciding" to "exceeding". Line 127: configuration "being" by far. Line 128: Dynamic flux "bags" (DFB) "have" been applied for flux "measurements". Line 134: Change sentence to "Reported DFC volumes and flushing flow rates range from". Line 135: resulting "in" a turnover. Line 136: "the" Hg0 flux. Line 138: "the" Hg0 flux. . . "the" DFC. . . "the" DFC. Line 140: Change "calculation" to "calculations". Line 142: Change "designed" to "design". Line 144: designed "DFCs". . . "and" showed that the. Line 147: "and" chamber dimensions. Line 148: "the" flushing flow rate. Line 151: transfer "has" indicated that "a" smaller. Line 152: "a" higher flushing. . . "a" higher measured. Line 153: "the" measured . . ."the" flushing. Line 154: when "the fluxes" obtained. Line 156: Change "condition" to "conditions". Line 157: Change "estimation" to estimations". Line 158: provide an "aerodynamically-designed" chamber. Line 160: allows "the utilization of the" ambient . . . calculate "the" flux. Line 162: Change "condition" to "conditions". . . is "the" overall. Line 165: Change "balance" to "balances" and "flux" to "fluxes". Line 166: "The" DFC flux. Line 167: Change "assumed" to "assumes" and "was" to "is". Line 168: Change "concentration" to "concentrations". Line 170: for
each "of the" calculated "fluxes". Line 174: replication "of" DFC. Line 178: and "the" spatial scale of flux "footprints" Line 179: Change "flux" to "fluxes". Line 180: Remove "a". Line 181: measurements "are" currently "comprised" of "the" relaxed... "the" aerodynamic. Line 182: "the" modified. Line 183: "is" a direct. Line 184: Is it possible to find another word for "realized"? Line 185 and 186: Change "measurement" to "measurements". Line 189: "sampling heights" not "heights sampling". Line 190: that "the" REA. Line 191: Remove ", which"... change "heights" to "height". Line 192: "samplings" in "using" gradient... introduced "through the" forming. Line 196: "The" REA...change "measurement" to "measurements". Line 198: "The" AGM method... change "saltmarsh" to "saltmarshes". Line 200: "The" MBR method. Line 201: Change "floor" to "floors". Line 203: "requirements" of micrometeorology "are" less. Line 207: Change "surface" to "surfaces". Line 209: Remove "the" from "advection of Hg0 from the". Line 211: gradient "fluxes resulting" in.. vertical "fluxes" at... downwind of "an' NH3. Line 213: in "the" Nevada. Line 214: Change "error" to "errors" and "flux" to "fluxes". Line 215: terms on "the" net... evaluated. "Multiple" heights. Line 217: at "a" high. Line 218: Change "flux" to "fluxes". Line 221: Change "occur" to "occurs". Line 222: Change "coefficient" to "coefficients". Line 223: Change "is" to "were" typically. Line 225: Change "flux" to "fluxes". Line 227: "the" summer. Line 228: that "the" AGM and MBR "methods" observed. Line 229: Change "is" to "was" not satisfactory. Line 234: Change "understand" to "understanding. Line 236: Change "flux" to "fluxes". Line 239: Change "flux" to "fluxes". Line 240: were not sufficient "in" "eliminating" the. Line 243: Although "the" MBR "methods showed" substantial. Line 246: upwind "from" the sampling. Line 250: "the" MM fluxes... "the" DFCs. Line 251: to those "for" temperature. Line 255: times "that" of. Line 256: reduced "the" uncertainty. Lines 257-258: MBR "fluxes" are weak because of "the" high variability "in the" MM "fluxes"... mean "fluxes" from simultaneous "measurements". Lines 264-265: factor of "approximately two"...result "of" the fact....utilized a "relatively" low. Line 266: underestimating "the" surface... "a" Mann-Whitney. Line 267: Change "Probability" to "Probabilities". Line 269: "as well as differences in the measurement sites and periods.". Line 270: of "the" MM methods.
[Figure]

Line 272: compared "to" DFC measurements "which" lasted. Line 276: "found that the observed median MM flux". Line 277: Change "flux" to "fluxes". Line 278: "suggested that an elevated flushing flow rate generated a partial vacuum inside the DFC and created an artificial HG0 flux from the soil". Line 280: "the" DFC. Line 281: Change "DFC" to "DFCs". Line 282: Change "difference" to differences". Line 283: "differences in the median fluxes.". Line 289: Change "of" to "in". Line 290: "the" formation..."the" mass transfer. Line 291: found "to be" highly. Lines 296-297: by "the" Arrhenius (both times)...explain "the" Hg0 flux. Line 298: implying "that" other. Line 299: "i.e. wind and surface friction". Line 300: factor "that drives" the Hg0 release. Line 306: Change "molecular" to "molecules" and "replaces" to "replace". Line 309: moisture, "a" maximum. Line 310: Change "become" to "became". Line 311: showed "a" smaller increase. Line 318: Change "less" to "lesser". Line 320: by reducing "the" Hg0. Line 322: Remove "the". Line 327: Change "interacts" to "interact". Line 335: Change "provide" to "providing". Line 336: Change "surface" to "surfaces". Line 338: Change "suggested" to "suggesting". Line 340: Change "debates" to debate". Line 347: Change "measurement" to "measurements". Line 350: source of Hg in "the" leaf. Line 355: Remove "on". Line 357: instance, "the" higher Hg concentration..."suggests". Line 361: content in "the" leaf. Line 363: Change "flux" to "fluxes". Line 368: "biologically assimilated Hg retained in the leaf". Line 370: from "the" leaf. Line 372: Change "drive" to "drives". Line 373: hypothesis of "a" compensation point. Line 374: Hg in "the" vegetation. Line 375: Change "mechanism" to mechanisms". Line 377: Hg by "the" leaf occurs... bonded in "the" leaf. Line 379: on "the" leaf ... finding is "that" the negative. Line 382: of "chemically-bonded" Hg in "the" leaf. Line 383: uptake by "the" plant. Line 387: "The" bulk method... Change "measurement" to "measurements". Line 388: approach for "the" oceanic. Line 389: which "is" generally. Lines 393-396: DGM is "the" dissolved... GEM is "the" near... HT is "the dimensionless" Henry's... is "the" wind speed... is "the" Hg0 diffusion...DGM in "the" water phase. Line 398: has "a" much higher. Line 399: by "the" water transfer.... Should Eq (5) be Eq (3)? Line 403: Change "controlled" to "controlling" and "regulated" to "regulates". Line 406: summarized "the" Hg... "Eq. (4)"

instead of 6? Line 407: Change "resembling" to "resembles". Line 411: Change "was" to "were". Line 415: Change "photo-oxidant" to "photo-oxidants". Line 427: Change "droplet" to "droplets". Line 435: production in "the" Baltic Sea. Line 437: is reduced in "the" cell's cytoplasm. Line 439: manganese as "a" terminal Lines 441-442: also "show capabilities in oxidizing" Hg0. Line 447: Change "arrays" to "array". Line 465: under "the" canopy. Line 478: studies "that" periodically... change "flux" to "fluxes". Line 479: fluxes of all "of the" campaigns "were" used. Line 481: Change "model" to "models". Line 482: Change "landscapes" to "landscape" and "into" to "to". Line 486: into "naturally enriched" and "anthropogenically contaminated" sites. Line 491: noted that "the fluxes" reported. Line 497: Change "investigation" to "investigations". Line 510: Change "of" to "or" and "statistics" to "statistic". Line 520: the "fluxes" over... results of "the" reemission. Line 524: Change "ranges" to "range". Line 531: Change "evasions" to "evasion". Line 538: Change "homogenized" to "homogeneous". Line 539: Change "measurement" to "measurements". Line 540: Change "was" to "were". Line 541: Hg0 "fluxes" observed in East Asia "are" consistently. Line 544: The "fluxes" over freshwater in Europe "are" somewhat. Line 550: Change "flux" to "fluxes". Lines 551 & 554: Change "mine" to "mines". Line 554: evasion "in the" daytime... pattern "for" most "of the" Earth surfaces. Line 556: uptake through "the" stomata. Line 557: during "the" daytime. Line 558: in "the" warm season... in "the" cold season. Lines 559-560: in "the"Adirondack... in "the" summer... in "the" winter. Line 564: Change "observation" to "observations" and "measurement" to "measurements". Lines 565-566: in "the" summer during "the" corn . . . uptake by "the" corn leaf. . . observed in "the" winter. Line 584: accuracy of "the" global... based on "the" empirical. Line 586: Change "uncertainty" to "uncertainties". Line 595: contributing to "the" regional . . . Hg0 "fluxes" from. Line 596: by using "the" LIDAR. Lines 599-600: "the" Alamden mine... "concentrations" and "fluxes" were estimated to be 0.1-5 "and" 600-1200. Line 603: Change "measurement" to "measurements" and remove "of flux". Line 604: Change "flux" to "fluxes". Line 606: "the" Idrijca River. Line 607: Change "flux" to "fluxes". Line 609: Change "is" to "are" and "areas" to "area". Line 615: Change "nearby" to "near". Line 619: Change

"system" to "systems" and "pack" to "packs". Line 623: Change "investigation" to "investigations". Line 627: Change "Flux" to "Fluxes" and "remains" to "remain". Line 628: Change "flux" to fluxes". Line 630: Change "source" to "sources" and "sink" to "sinks". Line 632: "have" suggested that "plants are" a net sink "for" atmospheric Hg. Line 633: Change "concentration" to "concentrations". Line 636: deposition, and "the" forest. Line 640: suggested "that the" forest. Lines 642-643: Change "observation" to "observations" and "is" to "are"...useful "in helping" to understand.... Change "forest" to "forests". Line 646: Change "was" to "is". Line 647: Change "parameterization" to "parameterizations" and "calculates" to "calculate". Line 648: that Hg "is" passed. Line 649: and "is then" transferred... Remove "in". Line 656: Change "flux" to "fluxes". Lines 659&661: Change "simulation" to "simulations". Line 667: Change "Change" to "Changing". Line 668: models with "a" compensation. Line 673: model-estimated "fluxes were" 1-2. Line 675: flux "models require". Line 676: Presently, "in the" bidirectional. Line 680: Change "in" to "as". Line 681: Change "cell" to "cells". Line 686: of "the" HgII. Line 695: in "the" presence of. Lines 696-697: Change "model" to "models" and "surface" to "surfaces". Line 700: Remove "in"...since "the" mid-1980s. "A" substantial. Line 704: Change "flux" to "fluxes" and "is" to "are". Line 706: "a" strong anthropogenic. Line 707: Change "constrains" to "constraints"...analysis of "the" atmospheric. Line 708: Change "measurement" to "measurements". Line 709: Remove "in"... Change "flux" to "fluxes". Line 710: Change "limited" to "limits". Lines 715-717: "sensitive sensors for determining Hg0 concentration gradients is of prime importance in improving flux data quality and in reducing uncertainties in the global assessment of the Hg budget.". Line 719: "establishment of a data comparison strategy". Line 720: large, "a" standardized. Line 721: Change "flux" to "fluxes"... "A" fundamental. Line 722: Change "flux" to "fluxes". Line 727: Change "response" to "responses". Line 728: defined in "a" statistical sense. Line 731: Change "analysis" to "analyses". Line 732: Change "flux" to "fluxes". Line 733: "Forests are most". Line 735: Change "database" to "databases". Line 737: Change "flux" to "fluxes". Line 738: providing "a" better. Line 740: Change "model" to "models". Line 744: Change "address" to "addressing". Line

745: Change "benefits" to "benefit" and "parameterization" to "parameterizations". Line 799: "Fig. 6" should be in bold. . . Change "flux" to "fluxes".

---

## Author Comment (AC1) · 25 Mar 2016

**Response to comments on "Global Observations and Modeling of Atmosphere-Surface Exchange of Elementary Mercury – A Critical Review" *by* W. Zhu et al.**

We thank the reviewers for their thoughtful and constructive comments that help improve the quality of our manuscript. We have incorporated the reviewers' suggestions in the revised manuscript. Our point-to-point response to the reviewers' comments are shown below.

**Anonymous Referee #1:**

Overall comments:

Overall this is a very well written and fully researched review paper. In addition to simply reviewing existing studies, this paper performs original analysis of the compiled datasets in order to make large scale observations. This paper provides a comprehensive assessment of the current understanding of the atmospheric surface exchange of Hg and recommend it for publication in Atmospheric Chemistry and Physics.

The only complicating factor with the publication of this paper is that a separate group published a fairly similar paper very recently: Agnan et al., New Constraints on Terrestrial Surface-Atmosphere Fluxes of Gaseous Elemental Mercury Using a Global Database. ES&T 2016.

Given the similarities in objective and scope, I think this paper by Zhu et al, needs to: 1) acknowledge this separate paper in results and discussion, and 2) specifically identify how their paper is unique from Agnan et al., 2016, and 3) discuss similarities and differences in the two papers findings.

*Response*: We deeply appreciate the reviewer for the supportive comments and constructive suggestions to our manuscript. We have recognized and review the paper by Agnan et al. (ES&T 2016); and agree with the reviewer that a more robust discussion pointing the specific contribution of this paper in addition to the paper by Agnan et al. The similarity of the two papers are mainly on the overlap on existing literature on Hg flux measurement. The major differences between the two papers are: (1) approaches in the data compilation and synthesis (e.g., the statistical treatments), (2) the coverage of flux data over different landuses (soil, forest, snow, freshwater, and ocean in this paper as compared to terrestrial surfaces in Agnan et al.), (3) the inclusion of mechanistic discussion on flux quantification approaches (e.g., enclosure and micromet measurements) and air-surface exchange processes (e.g., confounding influences by environmental factors), (4) the inclusion of flux modeling approaches and scale-up of flux

data for global cycle implications, and (5) the inclusion of more up-to-date field data and exclusion of laboratory data in the synthesis.

In the revised manuscript, we have provided an additional section to recognize the contribution by Agnan et al (2016) and laid out the differences of the two papers, cf. line 82-88. We have also cited Agnan et al. (2016) in other parts of our manuscript (line 284-293, line 498-501, line 603-605, line 648-649).

Specific comments:

Comment #1:     Line 261: Very interesting result. Glad to see this analysis. A little more information is needed. The samples sizes are 229 and 39, but it is not clear if these numbers represent daily average values, hourly values, etc. A little more discussion about what constitutes a measurement would be helpful. Also, within the <0.3 ug/g cutoff, were there significant differences in the Hg concentrations between DFC and MM areas? If not, this would help build the case for the analysis.

*Response*: We thank the reviewer for pointing this out. The information regarding the measurement conditions have been added (line 271-272). Because the total Hg concentrations in soil substrates are frequently not reported in the literatures, particularly in those studies from background sites, it is not reliable to compare the substrate Hg concentrations between DFC and MM measurements due to the small available sample sizes.

Comment #2:     Line 262: Why was 0.3 ug/g used as a cut-off point?

*Response*: The use of 0.3 $\mu$g g$^{-1}$ as the threshold of less human activity influenced background surfaces is based on investigation of background concentration from literatures and in line with the criteria used in Agnan et al. (2016). More important, Hg in a relatively low level surfaces are in general homogeneous than contaminated sites (Gustin et al., 1999), which reduced the uncertainty raised by footprint differences in comparing DFC and MM techniques.

Agnan, Y., Le Dantec, T., Moore, C. W., Edwards, G. C., and Obrist, D.: New constraints on terrestrial surface–atmosphere fluxes of gaseous elemental mercury using a global database, Environ. Sci. Technol., 50, 507-524, 2016.

Gustin, M. S., Lindberg, S., Marsik, F., Casimir, A., Ebinghaus, R., Edwards, G., Hubble-Fitzgerald, C., Kemp, R., Kock, H., Leonard, T., London, J., Majewski, M., Montecinos, C., Owens, J., Pilote, M., Poissant, L., Rasmussen, P., Schaedlich, F., Schneeberger, D., Schroeder, W., Sommar, J., Turner, R.,

Vette, A., Wallschlaeger, D., Xiao, Z., and Zhang, H.: Nevada STORMS project: Measurement of mercury emissions from naturally enriched surfaces, J. Geophys. Res.-Atmos., 104, 21831-21844, 1999.

Comment #3:  Line 288: change matters to matter.

*Response*: It has been changes accordingly.

Comment #4:  Line 285: Two factors that have been shown to affect soil-air Hg fluxes are grain size and soil disturbance. Only a couple of studies have shown this, but may want to consider including these two factors in the discussion if the goal is to be comprehensive as possible.

*Response*: We thank the reviewer for pointing out the two factors that are not as extensively studied: grain size and soil disturbance, and have provided the discussion in the revised manuscript, line 326-329.

Comment #5:  Line 339. There is a paper by Mazur et al. 2014 in Science of the Total Environment that has a similar focus: the impact of forestry operations on surface-air Hg fluxes.

*Response*: The results of Mazur et al. (2014) has been incorporated in the discussion and the reference has been added in the citation list, cf. line 348-349.

Comment #6:  Line 346: Suggest changing to "more recent" instead of just "recent". This idea has been around for more than a decade now.

*Response*: It has been changed.

Comment #7:  Line 352: remove excess Hg0.

*Response*: The excess $Hg^0$ has been deleted.

Comment #8:  Line 422: Need more information to support this statement. Earlier the text focuses on photo-pathways and this is a big jump without sufficient explanation.

*Response*: We agree with the reviewer on the suggestion. Previous statement was incorrect as showed in Fig.5, which has been corrected and reworded as "Both dark abiotic and biotic redox transformations are suggested to be involved (Fig. 5)", cf. line 435-436.

Comment #9:      Line 472: remove "got flux calculation".

*Response*: It has been removed.

Comment #10:     Line 476: in "the" literature

*Response*: It has been inserted into the text.

Comment #11:     Line 558: This paragraph should also discuss the work of Kuiken et al, 2008 part 1, which shows the opposite trend....lower emission in summer due to drier conditions and lower light from more leaf cover. In the scaling paper, Hartman et al, 2009 comes to the same conclusion.

*Response*: We thank the reviewer for pointing this out. The phenomenon of low flux in summer as a result of low light and drier conditions has been added in the discussion, cf. line 578-580.

Comment #12:     Line 601. Double check that Gustin et al, 2003 used a multivariate approach using soil Hg, flux and solar radiation. Or did that paper look at these variables separately.

*Response*: We thank the review for the cautionary remark. We have checked into Gustin et al. (2003) and discussed the influence of those environmental factors separately [page 345 and 347] with the citation.

*Comment* #13:     Line 700. Remove "in"

*Response*: It has been removed from the text.

*Comment* #14:     Line 713. This is a great summary of knowledge gaps, glad to see this in the paper.

*Response*: We thank the reviewer for the positive comment.

---

## Author Comment (AC2) · 25 Mar 2016

**Response to comments on "Global Observations and Modeling of Atmosphere-Surface Exchange of Elementary Mercury – A Critical Review" *by* W. Zhu et al.**

We thank the reviewers for their thoughtful and constructive comments that help improve the quality of our manuscript. We have incorporated the reviewers' suggestions and editorial corrections in the revised manuscript. Our point-to-point response to the reviewers' comments are shown below.

**Anonymous Referee #2:**

Overall comments:

This paper is a thorough review of measurement and modeling studies of elemental mercury. The depth and extent of the analyses of available data does indeed make this a critical review rather than just a literature review. The discussion on the advances in the measurement techniques is beneficial. This paper provides a necessary addition to the scientific community's GEM literature and aids in furthering our understanding of the air-surface exchange of atmospheric mercury. With some minor editing on a technical scale, I recommend the publication of this paper in Atmospheric Chemistry and Physics.

I agree with Reviewer #1's comments regarding the paper by Agnan et al. (2015). The only discussion on their paper was relating to the measurement method. It would be interesting to see a discussion on the findings of the two papers and how they complement each other.

*Response*: We deeply appreciate the reviewer for the supportive comments and constructive suggestions. Our special thanks to the reviewer for providing detailed editorial remarks. As discussed in our response to reviewer #1, we have recognized the review paper by Agnan et al. (ES&T 2016) and agree with the reviewer that a more robust discussion pointing the specific contribution of this paper in addition to the paper by Agnan et al. The similarity of the two papers are mainly on the overlap on existing literature on Hg flux measurement. The major differences between the two papers are: (1) approaches in the data compilation and synthesis (e.g., the statistical treatments), (2) the coverage of flux data over different landuses (soil, forest, snow, freshwater, and ocean in this paper as compared to terrestrial surfaces in Agnan et al.), (3) the inclusion of mechanistic discussion on flux quantification approaches (e.g., enclosure and micromet measurements) and air-surface exchange processes (e.g., confounding influence by environmental factors), (4) the inclusion of flux modeling approaches and scale-up of flux data for global cycle implications, and (5) the inclusion of more up-to-date field data and exclusion of laboratory data in the synthesis.

In the revised manuscript, we have provided an additional section to recognize the contribution by Agnan et al (2016) and laid out the differences of the two papers, cf. line 82-88. We have also cited Agnan et al. (2016) in other parts of our manuscript (line 284-293, line 498-501, line 603-605, line 648-649).

Specific comments:

Comment #1:     Line 108: Is this a possible typo that <1 Hz is considered a higher frequency?

*Response*: We thank the reviewer for pointing it out and would like to clarify it. The text has been rephrased as "Later on, monitoring ambient air Hg0 with relative higher frequency (up to 1 Hz) was achieved by using Lumex RA-915+ Zeeman atomic absorption spectrometry (AAS) analyzer operating without pre-concentration", cf. line 113-114.

Comment #2:     Line 113: The use of "but" in this sentence suggests that the higher detection limit of $0.35$ ng m$^{-3}$ is a negative aspect, but could that sensitivity be considered a benefit of this sensor over previous ones?

*Response*: Yes, the detection limit at $0.35$ mg m$^{-3}$ is a negative aspect in terms of analytical accuracy for Hg vapor because of the low concentration gradient typically observed in most flux quantification techniques (typically <0.4 ng m$^{-4}$, Zhu et al., 2015). However, the high frequency of this sensor (25 Hz) is a benefit among available Hg vapor detection techniques because it offers the possibility of using eddy covariance method to measure Hg flux even it is limited over contaminated surfaces only, e.g., Pierce et al., 2015.

Pierce, A. M., Moore, C. W., Wohlfahrt, G., Hörtnagl, L., Kljun, N., and Obrist, D.: Eddy covariance flux measurements of gaseous elemental mercury using cavity ring-down spectroscopy, Environ. Sci. Technol., 49, 1559-1568, 2015.

Zhu, W., Sommar, J., Lin, C. J., and Feng, X.: Mercury vapor air–surface exchange measured by collocated micrometeorological and enclosure methods - Part I: Data comparability and method characteristics, Atmos. Chem. Phys., 15, 685-702, 2015.

Comment #3:     Lines 185-187: This sentence, while accurate, discusses the lack of the ability of this sensor at background sites. This study however was over Hg-enriched soils and the sensor performed well over Hg-enriched sites. Would it be useful to note this as an advantage to this method considering the high number of sites that are Hg-enriched?

*Response*: We appreciate this insightful comment. It is indeed worth highlighting the possible application of CRDS-EC over Hg-enriched sites. The discussion has been revised to (cf. line 190-193):

Pierce et al. (2015) reported the first field trial of CRDS-EC flux measurement over Hg-enriched soils with a flux detection limit of 32 ng m$^{-2}$ h$^{-1}$, offered the opportunity for high frequently monitoring Hg$^0$ flux from Hg-enriched surfaces. However, the present state of development of CRDS-EC must be further advanced for Hg$^0$ flux measurement at most, if not all, background sites.

Comment #4: Line 547: Perhaps consider mentioning why the fluxes would be higher in Europe than East Asia prior to 2002 and during summer and/or daytime.

*Response*: We agree with the reviewer on the suggestion. Detailed discussion of high fluxes over freshwater bodies in summer and daytime have been discussed in Section 3.3 and 4.3.2. We have also revised the sentence to (cf. line 561-563) "The flux over freshwater bodies in Europe is somewhat higher than those measured in East Asia (6.5 vs. 4.6 ng m$^{-2}$ h$^{-1}$, *p*=0.40, ANOVA). These data were obtained mostly prior to 2002 (n=9) or during summer time and daytime (n=8) subject to higher blank larger extent of photo-reduction and evaporation."

Comment #5: Line 558: There are some studies that suggest the opposite (e.g. Lee et al., 2000; Fristche et al., 2008).

*Response*: We thank the reviewer for the suggestion, the discussion of opposite seasonal flux variation has been provided in the revised manuscript with the corresponding references, cf. line 578-582.

Comment #6: Lines 577; 593; 612; 627: The titles of the subsections in Section 4.4 include statements. Does this possibly change the flow of the paper?

*Response*: We agree with the reviewer that the full statements in the titles are somewhat distracting. In the revised manuscript, the subtitles have been incorporated into the text to maintain the flow of the text.